# Heat-assisted hot-hole transfer increases the surface-enhanced Raman activity of Au-TiO$_2$ nanoarrays

Mengya Zhang[1,7], Tongcheng Yu [1,2,7], Hao Liu [1,2], Chao Lin[1], Yaping Yang[3], Bowen Lv[1], Qi Zhang [1,4] ✉, Ming Chen [5] ✉, Tianshuai Wang [6], Weihong Hua[1,2] & Kai Han [1,2] ✉

Monitoring the evolution of molecules during photo and thermal synergistically induced physical and chemical processes is of paramount interest in fields including chemical, material, and energy research. Surface-enhanced Raman spectroscopy (SERS) is a highly promising technology in this regard, offering advantages of sensitivity, real-time, and label-free detection. However, the application of conventional SERS in high-temperature environments has faced challenges due to the inevitable loss of activity and decline in sensitivity. Herein, we synthesize Au-TiO$_2$ nanoarrays as SERS substrates, and an anomalous enhancement of Raman signal with increasing temperature is observed. The signal intensity increases by 11.41 times at 180 °C compared to that at 22 °C. This high-temperature enhancement in Raman activity is attributed to an underlying mechanism: heat-assisted hot-hole transfer, which enables 785 nm photon-induced hot-hole transfer from Au to TiO$_2$. Our work expands the application of the SERS technique for high-temperature chemical analysis and molecular diagnostics.

High temperature plays a key role in photocatalysis, as it can accelerate chemical reaction rates, improve energy conversion efficiency, and enable additional chemical reaction pathways[1–4]. Advanced plasmonic photocatalysis under high-temperature conditions combines photon and thermal effects, making it an effective method for optimizing chemical reactions[1–3,5]. Real-time observation of photocatalytic reactions at elevated temperatures is crucial for gaining a deeper understanding of the photo and thermal physical mechanisms in the catalytic reaction process. However, most existing in-situ techniques, such as scanning tunneling microscopy (STM), X-ray photoelectron spectroscopy (XPS), and Auger electron spectroscopy (AES), require stringent vacuum environments, rendering them unsuitable for analyzing real-world thermal processes. In contrast, surface-enhanced

Raman spectroscopy (SERS) can operate under ambient conditions, making it an appealing tool for the in-situ characterization of analyte transformations in complex environments[6–9]. Despite its potential, conventional SERS faces limitations at high temperatures, where the signal intensity typically diminishes or even disappears[9–13]. This poses a critical challenge in achieving reliable Raman spectroscopy performance under such conditions.

Pioneers have made considerable efforts to improve the stability and sensitivity of high-temperature SERS[9–13]. One approach involves surface coating with inorganic materials to protect nanosubstrates from thermal degradation, thereby ensuring the stability of SERS analysis[9–11]. However, this method often sacrifices the SERS enhancement effect, as the coating inevitably impedes charge transfer between

[1]College of Advanced Interdisciplinary Studies, National University of Defense Technology, Changsha, China. [2]Nanhu Laser Laboratory, National University of Defense Technology, Changsha, China. [3]College of Aerospace Science and Engineering, National University of Defense Technology, Changsha, China. [4]Department of Applied Physics, Nanjing University of Science and Technology, Nanjing, China. [5]School of Physics, Shandong University, Jinan, China. [6]School of Chemistry and Chemical Engineering, Northwestern Polytechnical University, Xi'an, China. [7]These authors contributed equally: Mengya Zhang, Tongcheng Yu. ✉e-mail: zhangqi24@njust.edu.cn; chenming@sdu.edu.cn; hankai0071@nudt.edu.cn

probe molecules and substrates, leading to relatively weak probe signals[9–11]. Alternatively, polymers have been tailored to optimize the localized surface plasmon resonance (LSPR) characteristics of certain plasmonic hybrid substrates through temperature-triggered volume-phase transitions, resulting in improved SERS signals at elevated temperatures[12–14]. However, this method is restricted by the thermal stability of the polymers and currently applicable only below 55 °C, limiting its applications under higher temperature conditions[12,13]. To date, developing an active and robust in-situ SERS sensing technique for high-temperature applications remains challenging. Plasmonic metal–semiconductor hybrid materials have garnered attention owing to their enhanced stability by interfacial interactions[15,16]. These materials offer a promising opportunity for designing stable and sensitive SERS substrates that can withstand high-temperature conditions.

Here, using synthesized Au-TiO₂ nanoarrays (NAs) as substrates and under 785 nm light excitation, we observe that SERS signals from molecules are enhanced by 11.41 times at 180 °C compared to those at room temperature (22 °C). The application temperature of this high temperature-induced SERS (TI-SERS) technique is over 100 °C higher than that in previous temperature-dependent SERS studies[12,13]. Experimental observations from time-resolved spectroscopy and electron paramagnetic resonance spectroscopy (EPR) revealed the near-infrared driven hot-hole transfer from Au to TiO₂, beyond the only occurrences of electron transfer. Environmental heating changes the electron distribution near the Fermi level, enabling the transfer of hot holes. This process benefits the transfer of hot electrons from Au to the molecule, thereby enhancing Raman scattering. By contrast, at room temperature, the transfer of hot holes to TiO₂ is impeded, leading to reduced hot-electron transfer. Moreover, our TI-SERS technique exhibits sensitivity, tunability, reproducibility, stability, and versatility, enabling real-time monitoring of the plasmon-activated photothermal catalytic reactions. Our method offers not only stable and sensitive molecular detection techniques for high-temperature applications but also provides insights into thermal plasmonic dynamics in heterojunction materials, which hold potential in the fields of photocatalysis, photovoltaics, and phototransduction.

## Results

### Synthesis and characterization of Au-TiO₂ NAs

The original TiO₂ NAs with a large specific surface area were synthesized via a hydrothermal method, followed by a 25 min photochemical growth of Au nanoparticles (NPs) with SERS activity on the TiO₂ surface (Fig. 1a). Scanning electron microscopy (SEM) revealed the typical microstructures of the as-prepared Au-TiO₂ NAs, where the TiO₂ NAs were fully covered by a multitude of interconnected, even Au NPs with an average diameter at ≈19 nm (Fig. 1b and Supplementary Fig. 1). The densely packed and interconnected Au NPs on the TiO₂ surface effectively promote LSPR effects because of the strong coupling between Au NPs[6,17]. High-resolution transmission electron microscopy (TEM) provided detailed structural insights into the interface between TiO₂ and Au (Fig. 1c). The extended and interconnected Au lattice fringes at the boundary region indicate that the Au NPs are already overgrown on TiO₂, rather than being physically mixed. Lattice-spacing measurements revealed distances of 0.236 nm, corresponding to the face-centered cubic structure of Au[18], and 0.316 nm, corresponding to the (110) plane of rutile TiO₂[19] (Fig. 1c and Supplementary Fig. 2). The ring-like concentric diffraction pattern observed in the selective area electron diffraction (SAED) analysis (Fig. 1d) confirms the polycrystalline nature of the sample, further supported by the X-ray diffraction (XRD) patterns shown in Fig. 1j[18–20]. High-angle annular dark-field scanning transmission electron microscopy (HAADF−STEM) and energy-dispersive X-ray spectroscopy (EDS) mapping (Fig. 1e−i) reveal a uniform distribution of Au, Ti, and O elements throughout the sample. The calculated relative atomic ratios of Au, Ti, and O are

≈8.3:30.7:61.0 (Supplementary Fig. 7d). The XRD pattern indicates that the diffraction peaks of TiO₂ NAs are following the rutile phase (JCPDS No. 21-1276, Fig. 1j), and the Raman signal of TiO₂ NAs also exhibits distinct characteristics of the rutile phase (Supplementary Fig. 3), collectively confirming the formation of TiO₂ NAs in rutile crystal structure. The (110) plane of the TiO₂ NAs is almost perpendicular to the substrate, thus is invisible in the XRD pattern (Fig. 1j, Supplementary Fig. 4). XPS analysis reveals changes in the chemical state and binding energy of the Au-TiO₂ NAs (Fig. 1k). The Au $4f$ peaks at 84.1 and 87.8 eV corresponding to Au $4f_{7/2}$ and Au $4f_{5/2}$ exhibit distinct shifts in binding energy compared with pristine Au NPs due to the interaction and charge transfer between Au and TiO₂, which simultaneously modulate the electronic structures of each component[20,21], also resulting in the shift of Ti $2p$ peak (Supplementary Fig. 5). It reveals the first signs of Schottky barrier formation at the Au-TiO₂ interface, which is beneficial for the transfer of photoexcited hot carriers at the interface and the chemical enhancement for SERS activity[22].

We further investigated the optical properties of the Au-TiO₂ NAs using absorption spectroscopy (Fig. 1l). TiO₂ NAs exhibit sharp absorption peak with a cutoff at ≈380 nm. Following the overgrowth of plasmonic Au NPs on TiO₂ NAs, the absorption can be significantly extended to the range of 500–1000 nm, which substantially improves the light utilization efficiency of Au-TiO₂ NAs. To better understand the state of Au in the samples, additional experiments were conducted by varying the duration of the light irradiation growth process. Characterization results, including SEM, EDS, absorption spectra, and SERS spectra, are presented in Supplementary Figs. 6–8 and Supplementary Note 1. The analysis reveals that Au-TiO₂ NAs grown under light irradiation for 25 min exhibited the highest light-harvesting efficiency and LSPR properties. These samples were subsequently selected for high temperature SERS tests, and detailed findings are provided in Supplementary Figs. 6–8.

### High-temperature-induced SERS activity

Exciting the nanoprobe in the near-infrared region (785 nm) effectively eliminates substrate fluorescence interference compared to excitation with visible light (532 and 633 nm)[23]. Figure 2a shows the in-situ temperature-dependent SERS experiments conducted at 22, 90, 120, 150, and 180 °C using a common aromatic dye analyte[24,25]: crystal violet (CV) molecules ($10^{-7}$ M) with an excitation wavelength of 785 nm (1.58 eV). The Raman signal of CV molecules increased as the temperature rose from 90 to 180 °C. For instance, the Raman peak at 1172 cm$^{-1}$ increased from ≈3236 counts s$^{-1}$ at 22 °C to 14716 counts s$^{-1}$ at 120 °C, and then to a maximum of 36916 counts s$^{-1}$ at 180 °C, indicating a dramatic enhancement in SERS activity by a factor of 11.41. However, excessive heating (210 °C in this study) caused structural damage to the nanostructures, resulting in diminished SERS activity (Supplementary Fig. 9). The magnification factors for key Raman peaks of CV molecules at different temperatures are summarized in Fig. 2b. The data indicate obvious variability in the enhancement among different characteristic peaks in the TI-SERS measurements, with some peaks amplified by as large as 11.41 times, while some peaks had a lower enhancement factor of only 2.84 times. This selective enhancement of different Raman peaks and enhancement magnitude of TI-SERS (Fig. 2b and Supplementary Table 1) indicate an improved chemical enhancement in TI-SERS[6,7,24]. Control experiments were conducted to determine the contribution of individual components in Au-TiO₂ NAs to the TI-SERS effect (Fig. 2c and Supplementary Fig. 10). No appreciable SERS enhancement was observed at 180 °C when Au NPs or TiO₂ NAs were tested independently, ruling out their individual contributions to the TI-SERS effect. Also, room-temperature SERS testing of the 180 °C-annealed Au-TiO₂ NAs showed no detectable enhancement (Supplementary Fig. 10). In contrast, TI-SERS activity was extensively observed using other synthesized Au-TiO₂ substrates with varying Au components (Supplementary Fig. 11 and Supplementary Note 2),

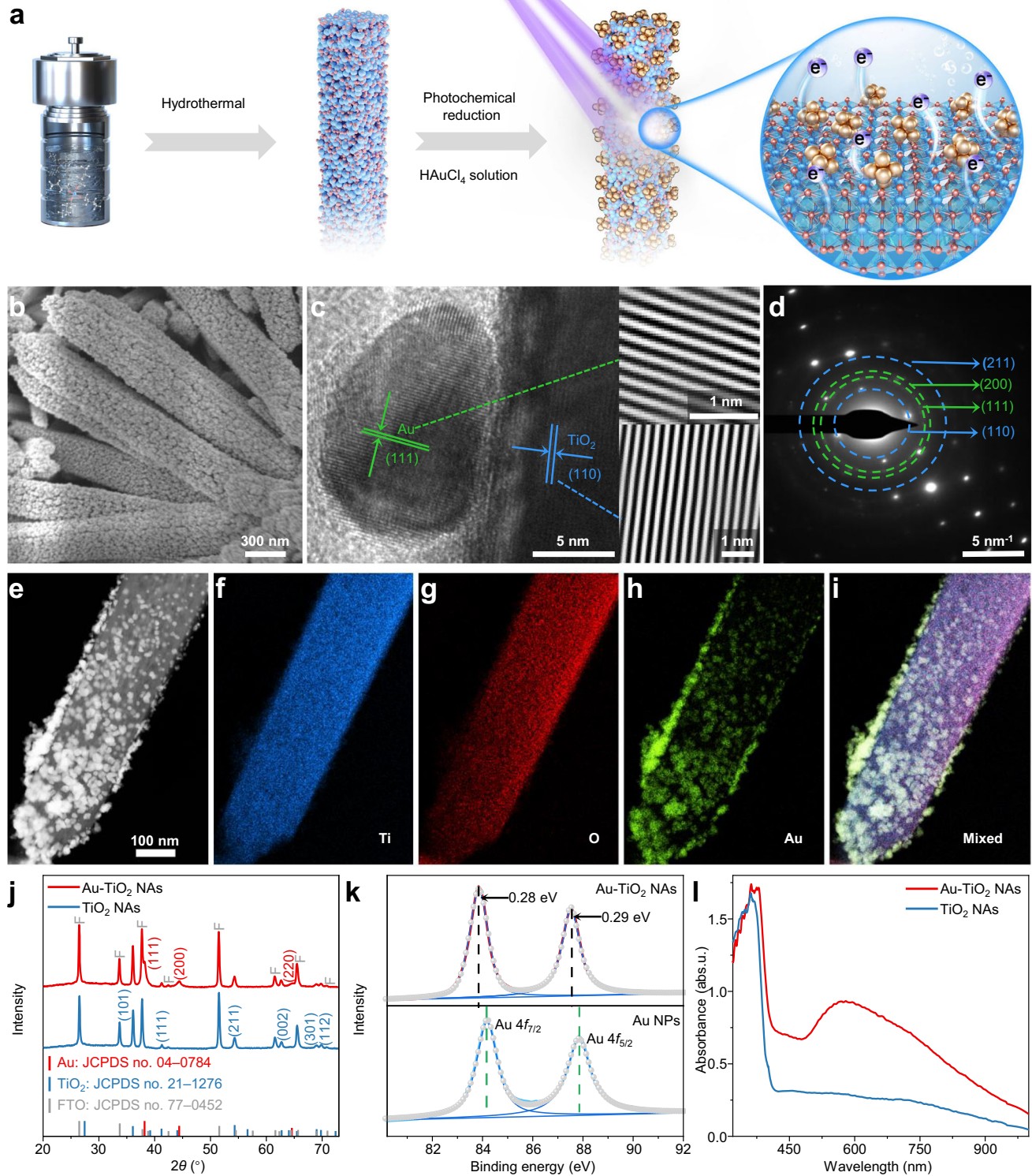

**Fig. 1 | Characterization of Au-TiO₂ NAs. a** Synthesis diagram of Au-TiO₂ NAs; Typical **b** SEM and **c** TEM images of as-prepared Au-TiO₂ NAs; **d** SAED pattern; **e–i** HAADF–STEM and EDS elemental mapping images of Au-TiO₂ NAs, and scale bar in **e** is valid for all panels from **e** to **i**; **j** XRD patterns of TiO₂ NAs and Au-TiO₂ NAs substrates; **k** XPS fine spectra of Au 4f originating from Au NPs and Au-TiO₂ NAs; **l** The absorption spectra of TiO₂ NAs and Au-TiO₂ NAs, and the unit (abs.u.) on the y-axis is equivalent to optical density. Source data are provided as a Source Data file.

suggesting that the TI-SERS effect arose from the synergistic interaction of Au and TiO₂ complexes under high-temperature conditions. Finally, CV molecules at concentrations ranging from $10^{-7}$ to $10^{-12}$ M were accurately measured (Fig. 2d and Supplementary Fig. 12), demonstrating the femtomolar-level detection capability of the TI-SERS technology.

## Tunability, reversibility, and stability

In addition to sensitivity, tunability, reversibility, and stability are also critical criteria for evaluating the practical performance of SERS substrates[9–11,26,27]. First, accurate control of the enhancement factor (EF), as a fundamental property of SERS, is necessary for quantitative detection[26]. As shown in Fig. 2e, the EF value clearly increased as the

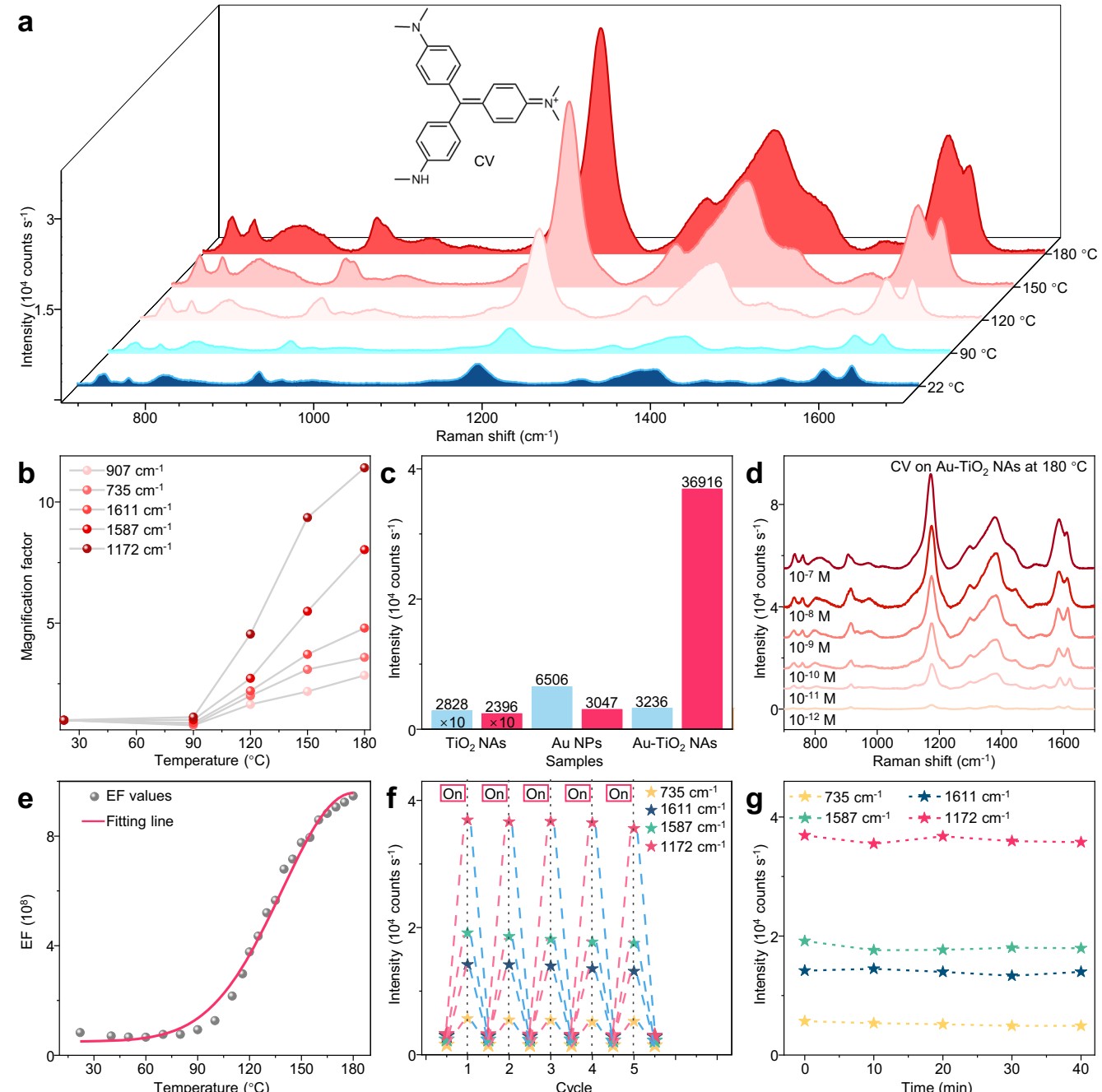

**Fig. 2 | TI-SERS spectra. a** TI-SERS spectra recorded from CV ($10^{-7}$ M) on Au-TiO$_2$ NAs with the temperature increasing from 22 to 180 °C (the inset shows the structures of the CV molecules); **b** The corresponding variation in the magnification factor with temperatures at several typical characteristic peaks of CV; **c** Raman signal intensities at 1172 cm$^{-1}$ for $10^{-3}$ M CV at TiO$_2$ NAs substrate, $10^{-5}$ M CV at Au NPs substrate, and the $10^{-7}$ M CV at Au-TiO$_2$ NAs substrate, recorded at 22 °C (blue bar) and 180 °C (red bar); **d** Raman spectra of CV molecules with concentrations of $10^{-7}$–$10^{-12}$ M; **e** The enhancement factor values of the characteristic peak at

1172 cm$^{-1}$ change with temperature, with the red line showing the signal change trend; **f** The variation in the Raman signal intensities (from several peaks) at a 180 °C thermal field in comparison with that at 22 °C during different cycles ("off" at 22 °C and "on" at 180 °C); **g** Raman signal intensities changing with time (0–40 min) for the Au-TiO$_2$ NAs substrate at 180 °C, the relative standard deviation of TI-SERS at 1172 cm$^{-1}$ within 40 min was only 1.7%. Source data are provided as a Source Data file.

temperature rose from 90 to 180 °C, and revealed a quantitative relationship (EF$\propto$e$^{-(T-180\,°C)^2}$, with $R^2$ of only 0.995) between the enhancement factor and temperature. The EF decay during cooling still followed a similar quantitative relationship (Supplementary Fig. 13), demonstrating accurate and controllable regulation of EF through temperature, which contributes to more precise SERS applications. Second, the reversibility of the substrates was confirmed through Raman intensity measurements during five consecutive

thermal field switching cycles (Fig. 2f). The enhanced Raman signals were well reproduced between 180 °C and room temperature with fluctuations of <4% (Fig. 2f and Supplementary Fig. 14), indicating the high reversibility of the substrates under the thermal field. Third, stability is among the most desirable features for high-performance SERS substrates in practical applications[9]. The Raman spectra can be clearly repeated even after 40 min of high-temperature treatment (Fig. 2g and Supplementary Figs. 15, 16). Moreover, the relative standard deviation

of the Raman peak intensities at 1172 cm$^{-1}$ was calculated to be ≈1.71%, highlighting the thermal stability of the TI-SERS system. In summary, the Au-TiO$_2$ NAs used as TI-SERS substrates exhibited sensitivity, tunability, reversibility, and stability, making them applicable for the analysis of analytes in high-temperature environments.

## Mechanisms underlying TI-SERS

The mechanism of action for TI-SERS remains unclear. On the one hand, some previous studies have shown that oxygen-related defects caused by high temperatures may enhance the Raman activity[24,28,29]. To verify whether TI-SERS is related to oxygen defects generated under high temperature conditions, we conducted a series of experiments, such as XPS, XRD, EPR, Raman and absorption spectra (Supplementary Figs. 17–21, 30, and Supplementary Notes 3, 4). The results indicate that, compared with room temperature, heating the Au-TiO$_2$ NAs substrates to 180 °C does not induce measurable changes in their structure or electronic configuration. This observation suggests that no oxygen defects are generated in the Au-TiO$_2$ NAs upon heating to 180 °C, indicating that the TI-SERS mechanism is not attributable to oxygen defects.

On the other hand, photo-induced charge transfer (PICT) plays a crucial role in SERS, as described by chemical enhancement theory[6,7,24,30]. Transient absorption spectroscopy (TAS) is widely used to investigate the charge transfer of materials[31,32]. Therefore, we employed TAS to investigate the carrier dynamics of Au-TiO$_2$ NAs. A 785-nm photon (1.58 eV) is used to excite the Au plasmon mode, but has no sufficient energy to excite TiO$_2$ (bandgap: 3.10 eV; Supplementary Figs. 21, 22). As displayed in Fig. 3a, b, photoexcitation of the Au NPs LSPR results in a winglet signal and a bleach signal (centered at 491 and 678 nm, respectively)[31,32]. The maximum amplitude of the winglet signal and bleach signal of Au considerably increased with increasing temperature (Fig. 3b), indicating a larger electron population on Au at high temperatures. As the charge transfer process of the Au nanoparticles reduces the electron–phonon coupling time[32–35] ($\tau_{e-ph}$), we analyze the carrier dynamics of Au NPs and Au-TiO$_2$ NAs at different temperatures. Signal fitting was performed using a double exponential decay convolved with the instrument response function. As illustrated in Fig. 3c, the dynamic process of Au NPs exhibited a $\tau_{e-ph}$ (8.6 ± 0.5 ps) at room temperature (bleach signal). At room temperature, Au-TiO$_2$ NAs (bleach signal with blue circles) exhibited faster $\tau_{e-ph}$ (6.7 ± 0.4 ps) than Au NPs, which has been previously attributed to the electron transfer process from Au to TiO$_2$[32]. While the $\tau_{e-ph}$ (5.2 ± 0.3 ps) of Au-TiO$_2$ NAs further decreases with increasing temperature to 180 °C. The mechanism underlying the shortened relaxation at higher temperatures remains unclear. Previous standard "solid-state" integral model[31,36,37] describes that the relaxation process slows down in metal with increasing temperature (laser heating), which is consistent with our experimental results (Supplementary Fig. 23). Moreover, the classic temperature-dependent permittivity models[38–41] agrees with the increased absorption we measured for Au NPs at 785 nm with rising temperature (Supplementary Fig. 24). Our data confirm that the relaxation time of Au NPs lengthened as the temperature increases (Supplementary Figs. 24, 25 and Supplementary Note 5). However, the shortened relaxation time observed in Au-TiO$_2$ NAs with increasing temperature (environmental heating) cannot be explained by the slowed relaxation predicted by the above models (Supplementary Figs. 24–26). The shorten $\tau_{e-ph}$ observed for Au-TiO$_2$ NAs at high environment temperature could be explained by the higher efficiency in hot carrier transfer, wherein accelerated relaxation via charge transfer aligns with previous experimental observations[31,32]. Similar fast relaxation dynamics trend was also obtained for the winglet signal decay (Supplementary Fig. 27). Furthermore, the selective chemical enhancement observed in our TI-SERS (11.41-fold at 1172 cm$^{-1}$, 2.84-fold at 907 cm$^{-1}$, Supplementary Table 1) also serves as evidence for efficient charge transfer, as opposed to a non-selective

enhancement[6,7,24,42,43] expected from purely electromagnetic mechanism. Next, direct evidence for hot-hole transfer was provided by our EPR observations.

In-situ EPR experiments were subsequently performed to further elucidate the roles of hot electrons or holes in the PICT process of Au-TiO$_2$ NAs at various temperatures (Fig. 3d and Supplementary Fig. 28). From the original EPR spectra (Supplementary Fig. 28), we have obtained the in-situ EPR light-minus-dark difference spectra (i.e., the EPR spectra under light irradiation at the same temperature minus those under dark conditions) for Au-TiO$_2$ at different temperatures, as shown in Fig. 3d. At relatively low temperatures (22–60 °C) under 785 nm light excitation, the EPR light-minus-dark difference spectra were dominated by signals of Ti$^{3+}$ ($g$ = 1.975, and 1.953 blue line) species, indicating that the transfer of photo-induced hot electrons from Au to TiO$_2$ resulted in the reduction of Ti$^{4+}$ to Ti$^{3+}$[44–46]. These results are consistent with previous findings at room temperature[44,47]. Moreover, under high temperatures (90–180 °C) combined with light irradiation, the new EPR-active O$^-$ species signals emerged at $g1$ = 2.003, $g2$ = 2.013, and $g3$ = 2.024 (denoted by h$^+$(O$^-$), red line), along with the increasing of Ti$^{3+}$ signals. This finding suggests the transfer of hot holes from Au to TiO$_2$[44–46]. Previous studies have confirmed that EPR-active O$^-$ species are indicative of hot-hole transfer from Au to interface state of TiO$_2$ under visible light excitation[44–46]. However, our results demonstrate that near-infrared light (785 nm) can induce hot-hole transfer from Au with the assistance of high temperature. These findings reveal a regime of interfacial energy transport. Furthermore, the EPR intensity of h$^+$(O$^-$) considerably increased with rising temperatures, reaching its highest value at 180 °C (red line). This indicates that high temperature not only facilitates but also enhances the hot-hole transfer process on Au under 785 nm excitation, highlighting the critical roles of thermal effects in expanding the scope of interfacial charge dynamics. Meanwhile, the computational investigations of Au-TiO$_2$ NAs indicate an increased probability of hot-hole transfer from Au at high temperatures, with detailed analysis provided in Supplementary Figs. 31, 32 and Supplementary Note 6. These computational results are consistent with the experimental findings. In addition, Ti$^{3+}$ signal also tended to increase as the temperature increased from 90 to 180 °C (Fig. 3d). We further confirmed that the decrease in Ti$^{3+}$ does not affect the performance of TI-SERS through the EPR and Raman results in an oxygen atmosphere (Supplementary Fig. 29), thereby ruling out the contribution of Ti$^{3+}$ to TI-SERS behavior. Moreover, we discussed the temperature-related quantum-tunneling effect[48–53]. The quantum-tunneling probability in plasmonic metal heterostructures is commonly considered to be weak dependence on temperature[50–53]. In our TI-SERS and EPR experiments, the signals are strongly dependent on temperature, indicating that the quantum-tunneling effect could be negligible.

Furthermore, density functional theory (DFT) calculations reveal that hot electrons of Au-TiO$_2$ NAs can be effectively transferred to CV molecules within the TI-SERS system (2.13 |e$^-$|, Fig. 3e and Supplementary Fig. 33). In general, extracting hot holes from Au NPs to semiconductors enables more charges to be available for chemical enhancement in SERS[24,34]. We proposed a chemical enhancement model dominated by PICT (Fig. 3f, Supplementary Figs. 34, 35, and Supplementary Notes 7, 8). Under high temperatures and 785 nm light excitation, hot holes on Au NPs can be efficiently transferred to TiO$_2$ (Fig. 3f, $P_{HT}$). This process facilitates charge separation and suppresses the recombination of hot electron–hole pairs in Au, thereby improving the transition efficiency of hot electrons from Au to the lowest unoccupied molecular orbital of CV molecules (Fig. 3f, $\mu_{PICT1}$). As a result, high-temperature conditions considerably enhance the PICT process between the Au-TiO$_2$ NAs and CV molecules under 785 nm excitation, which dramatically amplifies the molecular polarizability and the Raman scattering cross-section[6,7,54], thus enhancing the Raman signal of CV. However, as the temperature increased, no obvious TI-SERS

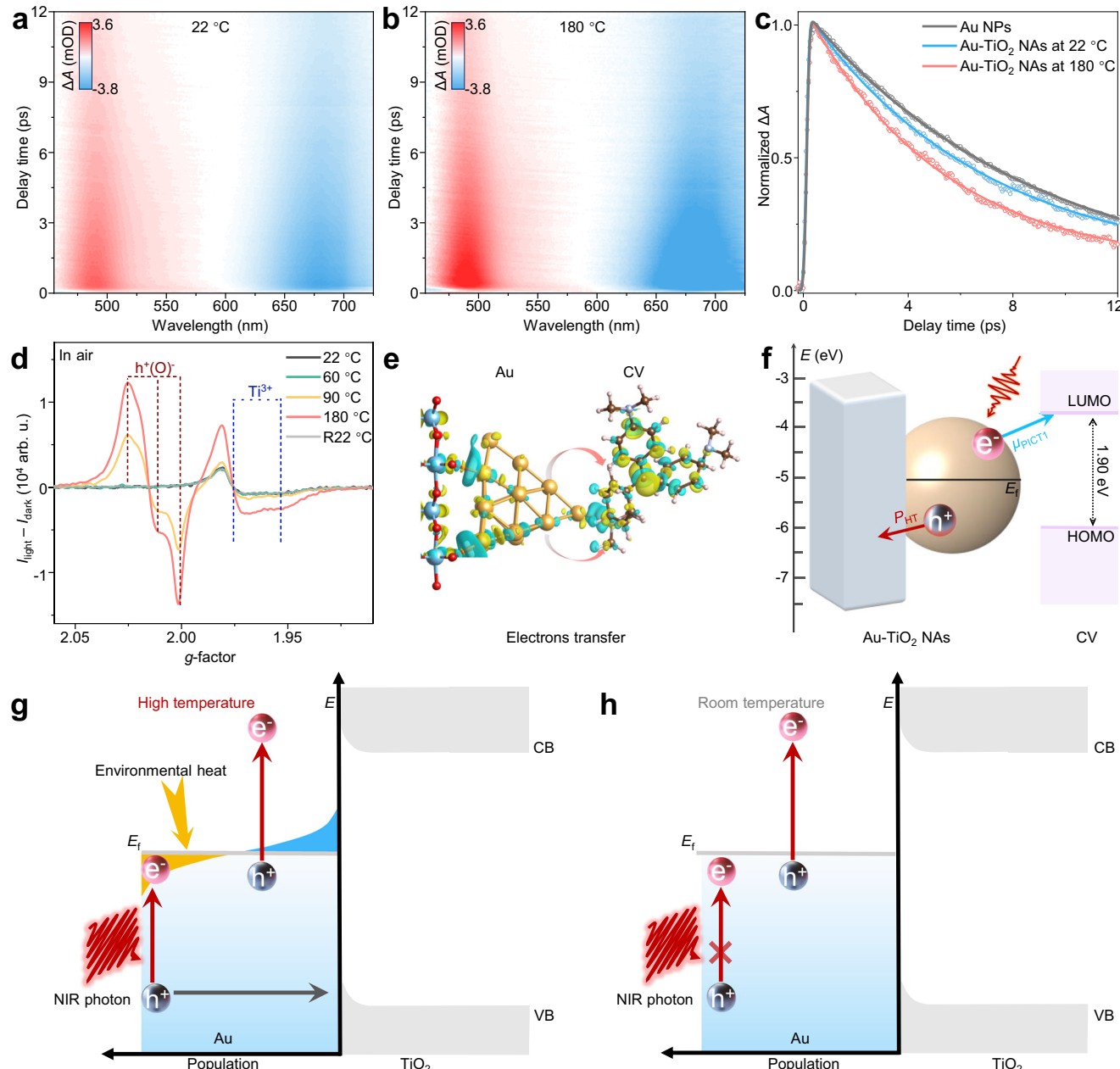

**Fig. 3 | Mechanism analysis of TI-SERS.** TAS map of Au-TiO₂ NAs obtained at **a** 22 °C and **b** 180 °C upon constant 785 nm pump fluence; **c** Dynamics of Au NPs at 22 °C, Au-TiO₂ NAs at 22 and 180 °C, extracted at the maximum bleach peak of Au; **d** In-situ EPR light-minus-dark difference spectra of Au-TiO₂ NAs at different temperatures under 785 nm laser irradiation, and R22 °C represents the in-situ cooling to 22 °C (the spectra were obtained in an air atmosphere); **e** Charge-density difference distribution of CV molecules with Au-TiO₂ NAs charged by additional electron; **f** Band alignment of Au-TiO₂ NAs with CV molecules under high temperature, where the vacuum level is considered as a reference (0 eV); Schematic diagram of hot holes transfer from Au to TiO₂ at **g** high temperature and **h** room temperature. Source data are provided as a Source Data file.

enhancement was observed when Raman lasers with excitation wavelengths of 532 and 633 nm (Supplementary Fig. 36) were used. Combining the above results, we conclude that high temperatures can trigger the photoexcitation of hot holes at higher energy levels, enabling their transfer. Specifically, the photon energy of 785 nm (1.58 eV) is slightly below the threshold energy (1.6 eV) required for hot-hole transfer at Au-TiO₂ interface[44,47]. However, works of pioneer have established that high temperatures could alter the electron distribution near the Fermi level[5,55]. This results in a consequent depletion of conduction band electrons near the Fermi level, thereby providing unoccupied states for the photoexcitation of hot holes at high energy levels (Fig. 3g). Therefore, Au-TiO₂ NAs can achieve efficient near-infrared-driven hot-hole transfer under high temperature. We term

this phenomenon as heat-assisted near-infrared driven hot-hole transfer, which is not permissible at room temperature (Fig. 3h).

## High-temperature SERS sensing and photocatalytic activity
The TI-SERS effect originates from the modulation of plasmon-generated hot carriers by temperature on Au-TiO₂ NA substrates. This phenomenon is not limited to specific probe molecules, highlighting its potential for broad molecular detection applications. To demonstrate the versatility of the TI-SERS effect, various molecules were tested, including a typical aromatic dye[24,25] methylene blue with large Raman cross-section, and a range of molecules with low Raman cross-section and weak affinity[24,56–58], such as glibenclamide, metformin hydrochloride, thiamphenicol, streptomycin, folic acid, thiram,

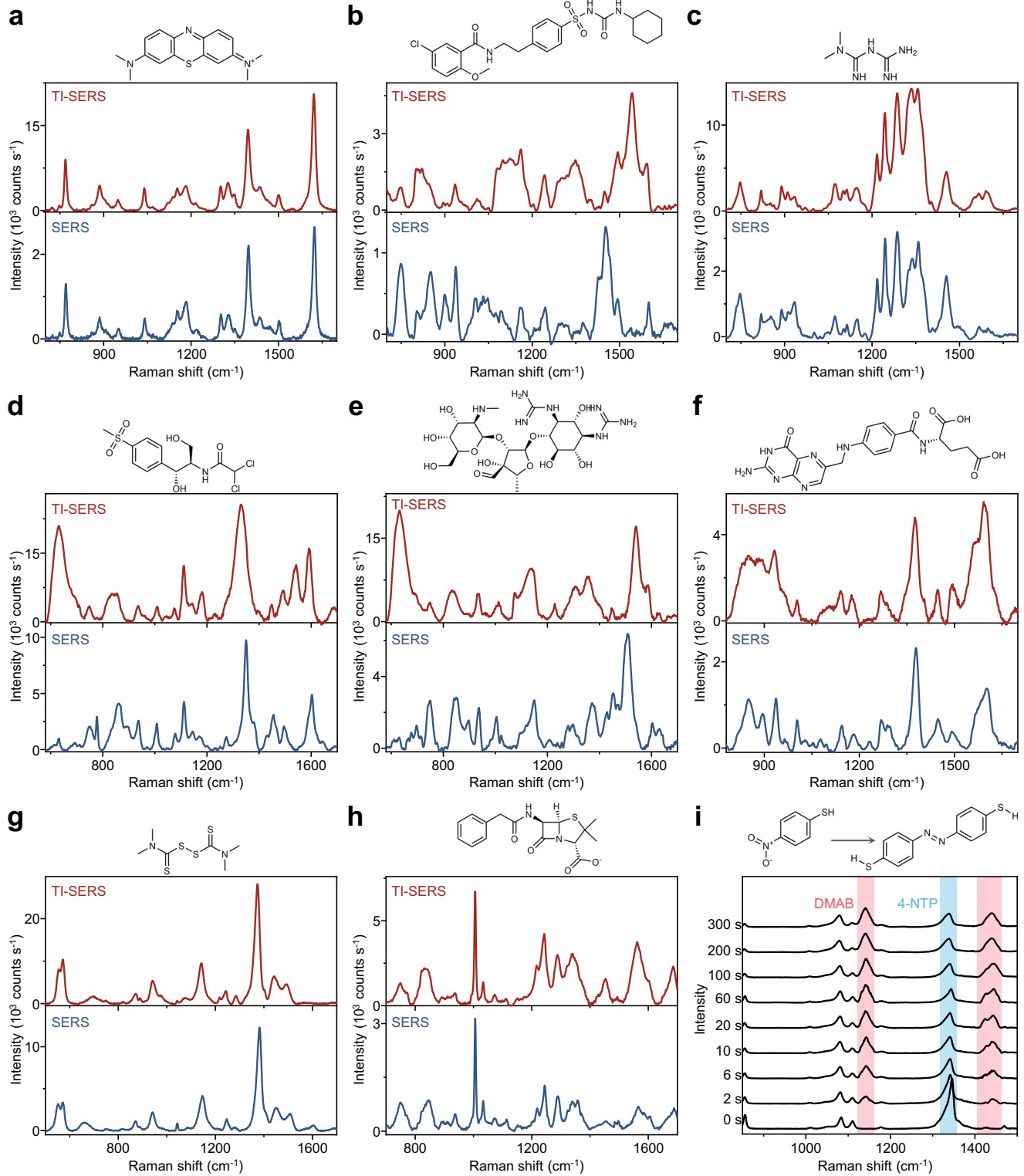

**Fig. 4 | TI-SERS analysis of diverse species and photocatalytic reactions.**
**a** Methylene blue (MB), **b** glibenclamide, **c** metformin hydrochloride,
**d** thiamphenicol, **e** streptomycin, **f** folic acid, **g** thiram, and **h** penicillin G sodium;
**i** Under 785 nm irradiation and at 180 °C, time-dependent SERS spectra were
recorded at different intervals to monitor the plasmon-driven dimerization of
4-NTP into DMAB on Au-TiO$_2$ NAs. All experimental parameters were kept identical
for both TI-SERS and SERS measurements, with the sole exception of the tem-
perature. Source data are provided as a Source Data file.

and penicillin G sodium (Fig. 4a–h, and Supplementary Table 1). The
results revealed considerable enhancement in their TI-SERS spectra.
Furthermore, Supplementary Fig. 37 and Supplementary Table 2
illustrate the SERS sensing of bisphenol S down to the nanomolar level,
a highly carcinogenic health hazard that is released in substantial

quantities during plastic decomposition at high temperatures[59,60],
demonstrating the practical applications and effectiveness of our TI-
SERS technology (Supplementary Note 9).

Our method efficiently integrates thermal energy into conven-
tional plasmonic photocatalytic reactions, leveraging the hot-carrier

properties of Au-TiO$_2$ NAs at high temperatures to enhance catalytic activity. On the basic of this principle, the synthesized Au-TiO$_2$ NAs were designed for plasmon-mediated interfacial catalysis and in-situ SERS monitoring simultaneously at 180 °C with 785 nm laser excitation (Fig. 4i). The time-dependent Raman spectra revealed the emergence and growth of DMAB characteristic peaks at 1143 and 1440 cm$^{-1}$, which was accompanied by a reduction in the PNTP characteristic peak at 1340 cm$^{-1}$, confirming the dimerization of PNTP into DMAB[61]. Interestingly, the dimerization reaction was initiated almost instantaneously (within 2 s) upon laser irradiation on the Au-TiO$_2$ NAs, which demonstrated that efficient hot-carrier transport between Au and TiO$_2$ under the combination of high-temperature and optical fields was conducive to improving catalytic performance. Control experiments (Supplementary Fig. 38) confirmed that neither plasmon excitation nor high temperature (180 °C) alone could trigger the catalytic reaction. Therefore, the synthesized Au-TiO$_2$ NAs at high temperatures can efficiently drive chemical reactions that cannot occur at room temperature, with a reaction rate calculated to be 0.096 s$^{-1}$ (Supplementary Fig. 39), which is superior to previously reported high-power plasmon-driven catalytic experiments[61,62]. Overall, our substrates with hot-carrier properties at high temperatures offer an alternative approach for plasmonic photocatalytic reactions in high temperatures.

## Discussion

In summary, we have demonstrated a TI-SERS strategy for detecting diverse analytes and monitoring plasmon-activated photothermal catalytic reactions at high temperatures. Unlike conventional SERS, which suffers from signal degradation as the temperature increases, the TI-SERS platform achieves anomalous signal enhancement of one order of magnitude at 180 °C. The mechanism of this enhancement can be described as follows. Environmental heating alters the electron distribution, leading to the formation of unoccupied states near the Fermi level, which realize the near-infrared photoexcitation of hot holes at high energy and facilitate their subsequent transfer from Au to TiO$_2$ (Fig. 3g). Unlike previous works focused on the transport of hot carriers (either electrons or holes) under visible-photon excitation[31,63], this work demonstrates hot-hole transfer from Au to a semiconductor under near-infrared excitation. Moreover, TI-SERS excited by near-infrared light offers advantages of an improved signal-to-noise ratio, as it can effectively eliminate substrate fluorescence interference compared with visible light excitation. EPR and TAS experiments revealed that the hot-hole transfer channel on Au nanoparticles can be initiated by altering the temperature, which facilitates the charge transfer resonance between the substrate and probe molecules. Our TI-SERS strategy can simultaneously ensure signal amplification, tunability, reproducibility, and substrate stability. Moreover, the strategy can be extended to detect various species with small Raman scattering cross-sections at high temperatures. These distinctive properties establish TI-SERS as a sensitive method for in-situ studies of plasmonic photothermal reactions, with applicability across various high-temperature analytical and catalytic processes.

Furthermore, plasmonic photocatalysis at high temperatures has emerged as a promising approach, leveraging the combined advantages of photochemical and thermochemical processes[1-4]. Two primary mechanisms are proposed: (1) plasmonic nanostructures absorb and convert light energy into heat, directly driving catalytic reactions, or (2) LSPR-induced hot carriers in plasmonic nanostructures interact with adsorbed molecules, lowering the energy barrier for reactions. However, the explanation of the latter photo and thermal cooperative catalytic mechanism remains unclear, as whether the temperature affects the reaction before or after carrier transfer is still uncertain. Our results suggest that, before photoexcitation, environmental heating changes the electron distribution near the Fermi level. Following photoexcitation, this distribution facilitates the excitation of hot holes at high energy levels.

## Methods

### Materials

Titanium butoxide (99.85%) and chloroauric acid (HAuCl$_4$·4H$_2$O, ≥47.8% Au basis) were purchased from China Sinopharm International (Shanghai) Co., Ltd. Hydrochloric acid (HCl, 99%) and ethanol (≥99.9%) were obtained from Macklin. CV (99%), bisphenol S (≥99%), and 4-nitrothiophenol (4-NTP, ≥95%) were purchased from Aladdin Chemistry Co., Ltd (Shanghai, China). In addition, poly-vinylpyrrolidone (PVP, $M_w$ = 40,000 g mol$^{-1}$) was obtained from TCI (Shanghai). F-doped SnO$_2$ (FTO) conductive glass was customized in Guluo. All the reagents were of analytical grade and used as received without further purification. Deionized (DI) water, used throughout the fabrication and measurement processes, was prepared using a Millipore purification system (18.2 MΩ cm).

### Synthesis of TiO$_2$ NAs

Self-supported TiO$_2$ NAs were synthesized on an FTO substrate using a hydrothermal method[19,20]. Initially, a 30.5 mL aqueous solution was prepared by mixing 15 mL of hydrochloric acid, 0.5 mL of titanium butoxide, and 15 mL of DI water in a glass dish, followed by stirring for 10 min to ensure homogeneity. The resulting mixture was then transferred to a 50 mL Teflon-lined autoclave. A precleaned FTO substrate was immersed in the prepared solution, and the hydrothermal reaction was conducted at 200 °C for 6 h. After the hydrothermal reaction, the sample was rinsed thoroughly with DI water. Finally, the self-supported TiO$_2$ NAs were obtained by subjecting the sample to heat treatment in air at 400 °C for 1 h.

### Synthesis of Au-TiO$_2$ NAs

The growth of dense Au particles on the TiO$_2$ NAs was achieved through ultraviolet (UV, 375 nm) laser irradiation[64], as shown in Fig. 1a. Briefly, the as-prepared TiO$_2$ substrate was immersed in a solution containing 20 μL of 0.05 M HAuCl$_4$, 0.5 mL ethanol, and 4 mL distilled water, then subjected to UV laser irradiation (wavelength of 375 nm, spot diameter of 2 cm, and power of ≈400 mW). The UV laser excited the TiO$_2$ semiconductor, generating a large number of electron−hole pairs on its surface. The active electrons effectively reduced Au$^{3+}$ to Au$^0$ to realize the growth of Au NPs on the TiO$_2$ surface. Simultaneously, the holes were rapidly consumed by the sacrificial reagent ethanol, which inhibited the recombination of photogenerated electron−hole pairs. This hole consumption also extends the electron−hole lifetime, promoting multielectron reduction of Au ions. Finally, Au-TiO$_2$ composite structures with varying Au contents were obtained by irradiation with a UV laser for different durations (0−45 min) and washing with DI water for further measurements.

### Synthesis of Au NPs

The Au NPs were synthesized through laser ablation using a Q-switched Nd-YAG (yttrium aluminum garnet) laser (Quanta Ray, Spectra-Physics). A 1064 nm pulsed laser beam (with a pulse width of 10 ns, energy of ≈250 mJ, and a repetition rate of 10 Hz) was employed to ablate the Au target in solution for 20 min, resulting in the formation of Au NPs. The as-synthesized Au NPs were then centrifuged at 6950 × g relative centrifugal force for 15 min, followed by resuspension in DI water for further use.

### SERS spectroscopy

SERS measurements were conducted using a confocal microprobe Raman spectrometer (Renishaw Raman spectroscopy) with a 50× objective. For a typical Raman spectroscopic analysis, the prepared substrates were individually immersed in 1 mL of analyte solutions (with ethanol as the solvent) and allowed to dry spontaneously for SERS measurements. Moreover, variable-temperature SERS tests for all the analytes were performed using accessories from the Linkam Scientific Instrument, and temperature increased at a constant rate of

10 °C min$^{-1}$. Upon reaching the desired temperature, the temperature was stabilized for 5 min to ensure experimental accuracy. All SERS spectra were excited using a 785 nm laser with an output power of 0.45 mW. The acquisition time for each spectrum was set to 1 s. Furthermore, measurements at different temperatures were performed at the same sample location to ensure comparability of the experiments. The SERS enhancement factors of analytes were calculated by utilizing the Eq. (1):

$$EF = (N_{NOR} \times I_{SERS})/(N_{SERS} \times I_{NOR}) \qquad (1)$$

Where $I_{SERS}$ and $I_{NOR}$ are the Raman intensities of SERS and Raman bands respectively ($I_{TI-SERS}$ represents the intensities of TI-SERS bands). The $N_{NOR}$, $N_{SERS}$, and $N_{TI-SERS}$ represent the corresponding number of analytes located in the laser spot area in the Raman, SERS, and TI-SERS measurements respectively, which can be approximately determined by the concentration of probe molecules[24,65]. It should be noted that all Raman measurements were carried out under identical experimental conditions. The analytes concentration of Raman were 0.2 M for CV, methylene blue, metformin hydrochloride, penicillin G sodium, streptomycin, folic acid, thiram, and 0.05 M for thiamphenicol, glibenclamide, respectively. The analytes concentration of SERS and TI-SERS were 10$^{-7}$ M for CV, methylene blue, 10$^{-6}$ M for glibenclamide, metformin hydrochloride, thiamphenicol, streptomycin, folic acid, thiram, and 10$^{-5}$ M for penicillin G sodium, respectively.

### Transient absorption spectroscopy
Femtosecond pulses at 800 nm with a pulse duration of 40 fs and a repetition rate of 1 kHz were generated by a Ti:sapphire oscillator-amplifier (Spectra-Physics). The femtosecond pulses were split into two beams: one directed into an optical parametric amplifier (Topas C, Light Conversion), and the other focused onto a CaF$_2$ or sapphire crystal to produce probe pulses. The generated probe pulses (a broad spectrum) were focused onto the sample and overlapped with the pump beam. In this study, 50 μL of test solution spun onto a glass substrate was activated by the optical parametric amplifier. Transient spectra were detected using a UV-NIR detector from a fiber spectrometer (AvaSpec-ULS2048CLEVO, Avantes). The sample was heated in-situ using a Cindbest temperature-controlled stage equipped with a platinum resistance thermometer, which established a stable high-temperature environment through ambient thermalization. The instrument response function (IRF) is 200 fs. The extracted kinetic traces at the probe wavelength were normalized for comparison, and electronic relaxation signals were deconvoluted by fitting with the Eq. (2)[32,66]:

$$S_e(t) = IRF \bigotimes \left( A \cdot e^{-\frac{t}{\tau_{e-ph}}} + B \cdot e^{-\frac{t}{\tau_{ph-ph}}} \right) \qquad (2)$$

where $S_e(t)$ is the electron signal intensity with time delay, and the symbol $\otimes$ denotes the convolution operation. $A$ and $B$, $\tau_{e-ph}$ and $\tau_{ph-ph}$ (phonon–phonon coupling time), are the amplitudes and time constants of the two exponential decay components.

### Two-temperature model
The two-temperature model[67–70] was used to simulate the relaxation phenomenon of Au NPs under different conditions, in which the energy exchange rate between electron and phonon are defined as Eqs. (3), and (4):

$$C_e \frac{dT_e}{dt} = -G(T_e - T_l) \qquad (3)$$

$$C_l \frac{dT_l}{dt} = G(T_e - T_l) \qquad (4)$$

Where $C_l$ (2.5×10$^6$ J m$^{-3}$ K$^{-1}$)[69] and $T_l$ are the heat capacity and temperature of lattice. And $C_e$ and $T_e$ are the heat capacity and temperature of electrons. For $T_e$ <3000 K, their relationship is characterized by a linear dependence[71,72], $C_e = \gamma T_e$, where $\gamma$ (68 J m$^{-3}$ K$^{-2}$)[69] is the electronic heat capacity constant. And $G$ (1.1×10$^{16}$ W m$^{-3}$ K$^{-1}$)[73] is the electron–phonon coupling constant of Au. Based on the absorbance (optical density, OD) of the Au nanoparticles (Supplementary Fig. 24) and 785 nm pump laser energy density ($I_{pump}$) at different ambient temperatures, the energy density absorbed by the Au NPs ($I_{absorb}$) was calculated by $I_{absorb} = I_{pump} \times (1-10^{-OD})$, yielding values of 160 μJ cm$^{-2}$ at 22 °C and 222 μJ cm$^{-2}$ at 180 °C with constant pump fluence. The corresponding peak electron temperatures were estimated to be 1563 K at 22 °C and 1864 K at 180 °C, which were then used in the two-temperature model to simulate the temporal evolution of electron temperature in Au NPs under different temperatures.

### First-principles calculations
The DFT calculations in this work were performed with a periodic slab model using the Vienna Ab Initio Simulation Package[74–76]. The Perdew-Burke-Ernzerhof (PBE) exchange-correlation functional[77] was applied with the generalized gradient approximation. The projector-augmented wave method[78,79] was used to describe the electron–ion interactions, and the cut-off energy for the plane-wave basis set was 400 eV. The total energy convergence criterion for electronic interactions were converged within 10$^{-5}$ eV. All the adsorption geometries were optimized using a force-based conjugate gradient algorithm. A 15 Å vacuum layer was added to all the slab models to prevent interactions between periodic images. The Bader charge was calculated using the Bader Charge Analysis module developed by Henkelman et al.[80,81]. The 1 × 1 × 1 $k$-point meshes were employed for the TiO$_2$, Au-TiO$_2$, and Au-TiO$_2$ and CV molecule surface systems. The adsorption energy ($E_{ad}$) is defined as Eq. (5):

$$E_{ad} = E_{total} - E_{slab} - E_{adsorbate} \qquad (5)$$

where $E_{total}$ denotes the total energy of the slab model with adsorbate atoms/molecules, $E_{slab}$ denotes the total energy of the slab system without adsorbate, and $E_{adsorbate}$ denotes the total energy of adsorbate atoms/molecules.

During the band calculation, the on-site Coulomb interactions of the Au 5$d$ and Ti 3$d$ electrons were treated within the Dudarev-type DFT + U formalism, with effective $U$ values of 1.5 eV for Au and 4.2 eV for Ti. Through this calculation, we obtained the fatband structures and density of states.

### Ultraviolet photoelectron spectroscopy
The energy band diagram of Au-TiO$_2$ NAs was calculated from ultraviolet photoelectron spectroscopy (UPS) results. The UPS results were obtained on Thermo ESCALAB XI+ instrument equipped with He-discharge lamp as the ultraviolet emission source (h$\nu$ = 21.22 eV). The work functions ($\Phi$) of TiO$_2$ NAs and Au-TiO$_2$ NAs can be determined by the Eq. (6):

$$\Phi = 21.22 - E_{cut-off} \qquad (6)$$

where 21.22 eV is the energy of the incident photon energy, and $E_{cut-off}$ is the cut-off energy of secondary electrons. Moreover, the extrapolated setoff energy ($E_{set-off}$) represents the difference between the Fermi level ($E_f$) and the valence band ($E_{VB}$). Based on these UPS results, we can evaluate the $\Phi$ or $E_f$, along with the $E_{VB}$ and conduction band ($E_{CB}$) positions within this system.

### Characterization
The morphologies and chemical compositions of the products were characterized using a focused ion beam electron microscope (Helios

G4 UC) equipped with EDS. TEM images and SAED patterns were obtained with a JEOL JEM-2100F microscope operating at an accelerating voltage of 200 kV. Crystalline information was collected on the basis of XRD patterns using a Rigaku Smart Lab 9 kW system with Cu $K_\alpha$ radiation ($\lambda = 0.15406$ nm). The surface chemical states and compositions of the samples were analyzed through XPS using a PHI Quantera SXM. UPS measurements were conducted using a Thermo ESCALAB XI+ instrument with the He I line serving as the ultraviolet emission source ($h\nu = 21.22$ eV). Absorption spectra were obtained using a UV-Vis-NIR spectrometer (20/30 PVTM micro-spectrophotometer), which was equipped with a Linkam Scientific Instrument Heating Cell Kit. The in-situ EPR investigation was performed by a Bruker A300 spectrometer operating at the X-band frequency, which was equipped with an in-situ digital temperature control systems ER4131VT and ER4141VT working in the temperature range of 100–600 K. The nanoarray sample was loaded into quartz glass tubes connected both to a high-vacuum pumping system and to a controlled gas feed (air or $O_2$). Irradiation was performed by a 785 nm laser (Leoptics) and the output radiation focused on the samples in the cavity by an optical fiber. The spectra were recorded before and after 10 min of 785 nm laser irradiation inside the EPR cavity at different temperatures.

## Data availability
The data that support the findings of this study are available from the corresponding authors upon request. Unprocessed raw data are provided as Supplementary Data 1. Source data are provided with this paper.

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

## Acknowledgements

The work was supported by the National Natural Science Foundation of China (22303107, Q.Z.;12104505, Y.Y.; and 11575102, M.C.).

## Author contributions

M.Y.Z., Q.Z., T.C.Y., M.C., and K.H. conceived the research and designed the experiments. M.Y.Z., Y.P.Y., C.L., B.W.L., and W.H.H. performed the synthesis and characterization of samples. M.Y.Z., T.C.Y., and M.C. performed the SERS and EPR experiments. H.L. performed the in-situ XPS experiments and in-situ EPR experiments. M.Y.Z., T.C.Y., and Q.Z. performed the ultrafast studies. C.L. contributed the theoretical calculations., T.S.W. performed the band structure calculations., M.Y.Z., T.C.Y., Q.Z., M.C., and W.H.H. analyzed the data. Q.Z., T.C.Y., H.L., and K.H. discussed the results. All the authors commented on the manuscript.

## Competing interests

The authors declare no competing interests.
