## [Transparent Peer Review file · Nature Communications]

Heat-assisted hot-hole transfer increases the surface-enhanced Raman activity of Au-TiO₂ nanoarrays

Corresponding Author: Professor Kai Han

Version 0:

Reviewer comments:

Reviewer #1

(Remarks to the Author)
see attached.

Reviewer #2

(Remarks to the Author)

Zhang et al. present a new SERS substrate, composed of photodeposited gold nanoparticles on TiO₂ nanorods. This substrate shows anomalous, and yet unobserved, high temperature SERS performance, with an optimum at 180°C. The substrate is used for the detection of various analytes, and the researchers attempt to explain the mode of action.

The observation of high temperature SERS is, without doubt, an important finding that warrants rapid and prestigious publication. Based on the data presented by the authors, I have no doubt that this effect occurs in a stable and reproducible manner, and that it is highly useful for the detection of very small amounts of analytes. Regarding the mechanism, though, I have doubts on the validity of the authors hypotheses, which are outlined in detail below. I think that a major revision is needed to confirm and sharpen the hypotheses.

(1) It is not clear which crystal phase dominates in the TiO₂ NRs. The authors compare the XRD pattern of their sample to the reference pattern of rutile, but the peaks do not match well. The peaks also do not match well with anatase. In addition, the small peaks near 26, 34, 38 and 52° are likely to result from the underlying FTO substrate, which should be clearly marked. The authors instead conclude the presence of rutile from the TEM lattice fringes, but is this the only option? Or is it possible that the tubes are a mix of rutile and anatase, or that they are not well crystalline at all?

(2) I have doubts about the accuracy of fit in Figure S17. All fits are located significantly above most measurement points. It is expected that by statistical error, in case of a good fit, about half of all measurement points are located above the fitting curve and the other half below it. This is not the case here. Can you please comment how the fit was obtained, and why it was done in this way? Furthermore, is a decrease by ~half a picosecond actually a "significant decrease", as the authors mention? Or would it be possible that, for example, different portions of the same sample show a similar deviation? In a similar manner, though the fit in Figure 4c looks much more reliable, is the difference in relaxation times extracted from the fit statistically significant?

(3) Figure S18 shows that a Ti³⁺ signal is already visible in the dark, which is easily possible, given that laser irradiation for gold deposition may have slightly (photo)reduced the material. Since all EPR spectra are plotted in arbitrary units, it is not possible for the reader to decide whether more Ti³⁺ is present under irradiation. This is additionally made more complicated by the differences in temperature between the two measurements. This makes the "proof" for hot electron transfer appear weak. In addition, can the authors cite an article why the peak just below g=2 is assumed to belong to the Ti³⁺ signal? This is unexpected, given that the signal in Figure S18 displays exactly the same shape, yet without the additional small signal below g=2.

(4) UV-Vis spectra must be re-recorded and displayed (raw data) in the Supplementary Information. The Tauc plot cannot be

performed, the data is not clear. There is neither a clear baseline nor a clear absorption edge. Strictly speaking, the Tauc plot method is not appropriate anyway for composites of crystalline materials (see Klein et al., Adv. Funct. Mater. 33 (2023) 2304523). The reviewer understands, though, that the method is applied in comparison with previous works. Still, the band gap energy is estimated from the intercept of the tangent to the absorption edge with the baseline of the spectrum (not a random zero line), and neither of the two features (absorption edge, baseline) is clearly visible, as outline above.

(5) Considering points 1 and 3 above, it seems possible that the unexpected properties do not arise predominantly from an absorption by gold, but by titania. It is well known that reduced titania is blue, and that it shows broad absorption across the red and infrared range. The color of titania would be obscured, however, by the purple color due to the deposited gold. The alternative hypothesis of an action of reduced titania might be ruled out, for example, by an oxidative treatment of the final (gold deposited) substrate, for example at 180°C (where it is stable) with high amounts of oxygen in the surrounding atmosphere. If the samples change color, and the SERS performance at 180° is lost or strongly attenuated, then the mechanism should potentially take into account a contribution of the reduced (and possibly amorphous) TiO₂.

Reviewer #3

(Remarks to the Author)

The paper outlines work on SERS in high-temperature environments. The authors reported that Au-TiO₂ nanoarrays showed Raman signal enhancement with an increase in temperature. With a signal intensity increase at 180°C compared to that at 22°C. The paper ascribes this effect to phonon-assisted hot-hole energy upconversion. The paper showing that the increases in SERS signal, is probe molecule sensitive (Table S1 supplementary information).

The substrate consists of quasi-oriented nanotubes+silver nano particles (see Fig 4 supplementary information). The authors reporting that the sample is polycrystalline. Ag nanoparticles are grown on the TiO₂ through UV (375 nm) laser irradiation.

TI-SERS (needs to be define, but presumably meaning temperature induced SERS) studies from room temp to 180 degree C where undertaken. Before the distribution of the SERS studies the paper outlines characterization of the sample. Can the Authors address the following

Can information using methods be applied to determine in the impact of heating the substrate (upto 200 degrees), with regard to structure and electronic configuration (as seen via methods like XPS). Supplementary Fig. 7 shows that heating at c.a. 210 degrees is reported to cause melting of the sample. A very important point of information is to clearly assess the impact of heating to 180 degrees on the sample. Does this cause any persevered changes noting its near the melting point.

The use of UV light on TiO₂ has been reported to cause the PIERS effect, similarly the heating of semiconductors like TiO₂ also causes such effects as shown via heating semiconductors including TiO₂ to c.a. 200 degrees (see Nature Communications volume 7, Article number: 12189 (2016), Journal of Physical Chemistry C 124 (46), 25351-25360 2020 or ACS Applied Nano Materials 3 (2), 1666-1673, 2020). Significantly this extends to enhancing plasmon assisted catalysis. The author in Fig 5 showing that heating the substrate supports catalysis reactivity. Reports have shown that heating Ag/TiO₂ and other semiconductors (Journal of Physical Chemistry C 124 (46)) supports not only an increase in SERS but also in catalysis.

Overall. The noteworthy results are a demonstration that SERS at high temperatures in Ag/TiO₂ systems can be improved, by an amount depending upon the probe molecule. More work in regard to assessing the impact of thermal treatment of the substrate is requested. Such as taking the substrate and examining the improvements in SERS at room temperature using the oven treated substrate (as outlined in Journal of Physical Chemistry C 124 (46), 25351-25360 2020 or ACS Applied Nano Materials 3 (2), 1666-1673, 2020). A careful analysis of the Ag/TiO₂ substrate following heat treatment is required. Eg after heating at room temperature what is the ESR data and what are the changes in structure, if any. Does the presence of thermally induced defect states explain the observed ESR data for example.

Version 1:

Reviewer comments:

Reviewer #1

(Remarks to the Author)

The authors have answered the comments of the reviewers professionally, and improved their manuscript. I nevertheless still think that their proposed mechanism (now changed!) is not substantiated with existing theoretical knowledge. Specifically,

1. The authors indicate that in Au samples, the e-ph energy transfer rate slows down upon absorption/increase of temperature in their measurements and in previous work.
 - a. They mention changes to absorption upon, but it is not clear why is this relevant.
 - b. They do not explain the physical reason for the slowing down. I believe it is due to the increase of the heat capacity of the

electrons with temperature. This is something that should be easy to quantify by using known temperature dependence of the electron heat capacity and the temperature change.

2. Then, the authors contrast their finding of a faster e-ph energy transfer rate in Au-TiO₂ samples. Assuming the heat capacity of electrons in that sample is the same, they need to suggest a stronger effect. They suggested that "thermal excitation induces the depletion of conduction band electrons near the Fermi level, thereby creating unoccupied states that facilitate the excitation of hot holes from higher energy levels." – From Fig. 4g, I suppose they mean instead something like - excitation of hot holes at energy levels deep below the Fermi energy of Au -. Either way, such a mechanism was suggested in the context of plasmonic catalysis, and is controversial; indeed famous studies of the Halas group using this combination of materials has already been shown to raise incorrect claims due to lack or erroneous control experiments. Therefore, the authors should strengthen their (now fairly speculative) claim by quantifying it. Specifically, one should be able to predict the change of occupation at the hole state and near the Fermi energy by using the formulation suggested already in the 1970's by Rosei et al.. Since the temperatures and temperature change are both much smaller than the Fermi energy of Au, I expect from earlier studies of transient permittivity studies (where heating can be more than 10 times stronger) that the impact of the hole population change to be much weaker than the e-ph changes observed for Au only samples by orders of magnitude, especially since the transitions the authors seem to be discussing are near the X point. These are known to contribute much less than transitions near the L point. A possible exception may have been reported in the context of photoluminescence, but if so, should come with a strong justification. Moreover, the authors should be able to quantify not only the change in the number of thermal/non-thermal holes, but also their tunnelling probability to the TiO₂, and then, relate the process to the experimental observations of the SERS changes. All these steps are not trivial, and I am pessimistic they may yield convincing conclusions... but the authors deserve a chance to demonstrate. some related work was done by the group of Andrea Baldi.

I therefore suggest to make a decision about the suitability of the manuscript to Nature comm. Only after another revision.

Reviewer #2

(Remarks to the Author)

The article is a revised version that I had reviewed previously. The authors have submitted a rebuttal.

Indeed, the new work performed by the authors significantly increases the quality of the article. It seems now almost ready for publication, but a minor revision to clarify some additional questions is strongly advised:

(1) Why is the intense peak of the reference pattern of rutile (~27°, (100)peak) not observed in the diffraction pattern of the authors' sample? Does it have to do with the growth direction?

(2) Has a similar amount of sample, or even better, the exact same portion of the sample, been used in the EPR measurements? Otherwise, they are not quantitatively comparable.

Everything else is now clear, and I very much like to thank the authors for their effort.

Reviewer #3

(Remarks to the Author)

The paper reports a novel high-temperature enhanced SERS (TI-SERS) effect using Au-TiO₂ nanorod arrays (NRs) under 785 nm excitation. Unlike traditional SERS, which usually weakens at elevated temperatures, this study shows a >11 × enhancement at 180 °C. The authors attribute this to a heat-assisted hot-hole transfer mechanism from Au to TiO₂, which promotes hot-electron transfer to analyte molecules and thus enhances Raman scattering.

The authors have responded in detail to the list of questions/issues from the reviewers.

Overall I am happy that the paper be published. Noting the comments below.

I would like to point out that the manuscript is technically sound but occasionally verbose and could benefit from editing for clarity. For example:

"Anomalous observed Raman signal enhancement" → "An anomalous enhancement of Raman signal was observed."

Consistency in units (°C spacing, scientific notation, etc.) should be improved.

Version 2:

Reviewer comments:

Reviewer #1

(Remarks to the Author)

I read the response to previous round of comments. I find the answer of the authors confusing, and reflecting a lack of

understanding of relevant basic concepts. I am either not able to follow their logic, or do not find it convincing qualitatively/quantitatively. Due to that, at this stage, I recommend to reject the paper.

Specifically,

1. It is very unlikely that the absorption spectrum changed from 160 to 220 $\mu\text{J}/\text{cm}^2$ (upon heating from 300K to 450K) which is ~40%. Indeed, the thermo-derivatives of the permittivity (and hence the absorption) were measured by several groups to be on the order of $1\text{e-}3 - 1\text{e-}2 / \text{K}$ (see works by Shalaev group, Chu group and others).
2. I do not understand what is the meaning of the statement – “the electron-phonon relaxation prolonged by 1.5 ps at 180 °C compared to 22 °C”. what is the initial e-ph coupling time the authors have in mind? I would expect a change of decay time to be reported in percent. The value reported reflects an extremely large, unrealistic change, and raises an alarm about the whole argument. Does it even correlate to the experimental finding?
3. Either way, quantifying the decay (cooling) rate should not be related to the absorption stage; thus, referring to it is quite surprising, and raises doubt about the whole line of thought of the authors, especially since the equations 2-3 in the SI do not involve absorption...
4. Instead, I would have expected the authors to perform at least a crude estimate of the e-ph energy transfer time. A simple analysis reveals that what matters is not the heating level they refer to (which is similar for both cases, assuming the absorption does not change drastically), but rather the temperature itself. Such estimate should then be compared to the experimental measurement. No such analysis was done.
5. The authors write – “we quantified the change of the Fermi surface at 180 °C compared to 22 °C as 0.06 eV (Supplementary Note 7).” This is unlikely - this is 3 times the temperature...
6. The authors refer to a transition of “1.60 eV at the Γ point (Supplementary Figs. 31)”. This transition is irrelevant for interband transitions in Au (because the Gamma point has many occupied states in the conduction band), and implies the authors did not understand or did not read the earlier work by Rosei, which is a well accepted benchmark.
7. The authors write – “Therefore, 785 nm laser (1.58 eV) with the aid of 0.06 eV thermal energy could ...” – this reflects a lack of understanding of how photon absorption works and generates non-thermal holes.

Reviewer #2

(Remarks to the Author)

In their second revision, the authors have answered my queries and clarified all open questions in full. Publication of the article is now recommended.

Version 3:

Reviewer comments:

Reviewer #1

(Remarks to the Author)

I find the answers to the last review satisfactory. I am not convinced that all the provided answers are included in the revision. I encourage the authors to do that, as this strengthens their arguments.

Response to reviewer's comments

Reviewer 1 (Remarks to the Author):

The authors suggest Au-TiO₂ particles as a platform for high temperature SERS. They claim an enhancement of about an order of magnitude at about 100 degrees higher than before in a stable manner. The results may be technically useful, but I am not convinced by the mechanism proposed, since simpler explanations were not ruled out.

Author Reply: Thank you to the reviewer for the valuable comments and for emphasizing the importance of considering alternative mechanisms. We agree that simpler explanations should be examined and will include additional discussion in the revised manuscript (See Q2). We appreciate your insights, which help improve the clarity and robustness of our interpretation.

Q1. In particular, one of the main findings (based on TAS) is a reduction of the e-ph decay time between 4.3 ps to 3.7 ps upon heating to 180°C. This effect of 14% can be easily explained via a standard “solid-state” integral model of the effect, see, e.g., del Fatti et al., PRB 2000 (and countless re-uses of the approach for pulsed and CW illumination). There is also various simpler approaches (e.g., Grua et al., PRB 2003, Saavedra et al, ACS Photonics 2016). The effect may also be reflected via temperature-dependent permittivity models, e.g., by the Shalaev group (various ellipsometry studies for Au, Ag, ...), the Chu group (Scientific Reports 2016, Nanophotonics 2017, Opt. Exp. 2017), Gurwich et al. (PRE 2017), Ferrara et al. (Phys Rev Mat 2019), etc. Such a simple electromagnetism-based explanations should be accompanied by a quantification of the SERS enhancement, a procedure which is well established (e.g., Greg Sun and Khurgin, or more sophisticated models that properly account for spatial variations of the field enhancement around the plasmonic structure.)

Author Reply: We appreciate the reviewer's insightful comments. Based on the previous well-established standard “solid-state” integral model and experiments¹⁻³, earlier studies have shown that upon photon absorption, the electron temperature in noble metals rises rapidly and subsequently transfers energy to the lattice through electron-phonon coupling (Laser heating process). As reported in Laser heating studies, the measured electron-phonon relaxation process slows down with increasing pump fluence¹⁻³(Fatti et al., PRB, 2000, 61(24), 16956; Saavedra et al., ACS Photonics, 2016, 3(9), 1637; Tagliabue et al., Nat. Mater., 2020, 19(12), 1312), which can also replicate in our material, as shown in Supplementary Fig. 22. In contrast, in our experiment, the sample was heated in situ using the Cindbest temperature-controlled stage equipped with a platinum resistance thermometer (main lines 381–383). This configuration established a stable high-temperature environment, directly affecting the lattice temperature through ambient thermalization (Environmental heating process). For environmental heating process, the results show that for Au nanoparticles, when the temperature is raised from 22°C to 180°C, the relaxation dynamics (Supplementary Fig. 24) also slow down with increasing temperature, consistent with previous observations under laser heating¹⁻³ (Supplementary Fig. 22), suggesting that Au nanoparticles may conform to the standard “solid-state” integral model. However, in the Au-TiO₂ system, an anomalous behavior occurs: as the temperature rises from 22°C to 180°C, the relaxation dynamics (Fig. 4c) become faster, a phenomenon that cannot be explained by existing models. Next, in order to investigate the temperature-dependent permittivity of Au-TiO₂, we performed in situ steady-state temperature-dependent absorption measurements (Supplementary Fig. 23). Compared to 22°C, the

absorption of Au nanoparticles at 785 nm increases by $\sim 40\%$ at 180°C (Supplementary Fig. 23c), indicating that the rise in temperature significantly alters the dielectric constant of Au nanoparticles. The observation of predecessors¹⁻³ and our experimental measurements all indicate when a noble metal absorbs more photons, electron relaxation should actually slow down (Supplementary Fig. 24). However, the absorption spectra of Au-TiO₂ NRs increases by $\sim 1\%$ at 180°C compared to 22°C (Supplementary Fig. 23d), which cannot explain the faster decay of the dynamics of Au-TiO₂ NRs (Fig. 4c and Supplementary Fig. 25, main lines 193–205).

On the other hand, according to the temperature-dependent permittivity model⁴⁻⁷, the increase in the imaginary part of the dielectric constant at elevated temperatures leads to a reduction of the electromagnetic field⁴⁻⁶ (Response Fig. 1, Shalaev et al., ACS Photonics, 2017, 4(5), 1083; Chu et al. Opt. Exp., 2017, 24(17), 19254; Gurwich et al., PRE, 2017, 96(1), 012212), thereby reducing the SERS enhancement. The model is consistent with the experimentally observed decrease in SERS performance of Au NPs at 180°C compared to 22°C (Supplementary Fig. 9a), but contradicts with the enhanced SERS activity on Au-TiO₂ NRs under the same thermal conditions (Fig. 3a). Therefore, the electromagnetic explanations based on the temperature-dependent permittivity model fails to account for the TI-SERS enhancement of Au-TiO₂ NRs. It is well known that the electromagnetic enhancement mechanism in SERS provides uniform enhancement across all Raman peaks⁸⁻¹⁰ (Greg Sun and Khurgin, PRB, 2011, 84(4), 045415); Campion et al., Chem. Soc. Rev., 1998, 27(4), 241; Zhao et al., J. Raman Spectrosc., 2018, 49(8), 1257), whereas our TI-SERS system exhibits a mode-selective enhancement compared to SERS at room temperature, consistent with the signal amplification of specific vibrational modes dominated by charge transfer¹¹⁻¹³ (Chemical enhancement mechanism, Supplementary Table 1). The corresponding discussion is added in the main lines 193–209.

Response Fig. 1: The variations in **a** dielectric constant and **b** numerical simulation of field enhancement of single crystalline silver films (solid lines), along with electron beam evaporated polycrystalline silver films (dashed lines) at higher temperatures⁴; **c** Temperature-dependent permittivity at selected wavelengths with unannealed gold films:

the imaginary part ϵ'' ⁵; **d** The Lorentz-Drude fitting (symbols) compared to experimental result (lines): the imaginary part ϵ'' with the unannealed gold film⁵; **e** The temperature rise inside a Au nanosphere in an oil host illuminated at $\lambda = 550$ nm as a function of the incident intensity (black solid line), the solutions accounting for up to third, second, and first-order terms are shown by magenta dashed, red dash-dotted, and blue dotted lines, respectively⁶; **f** Imaginary parts of the Au permittivity as a function of the incoming intensity⁶, based on the various solutions for ΔT shown in **e**⁶; **g** Same as in **f** for the normalized electric field within the Au nanosphere⁶.

Revision: 1) We added this part in the main lines 193–209: “The mechanism underlying the shortened relaxation at higher temperatures remains unclear. The previous standard “solid-state” integral model¹⁻³ describes that the relaxation process slows down in metal with increasing temperature (Laser heating), which is consistent with our experimental results (Supplementary Fig. 22). Moreover, the classic temperature-dependent permittivity models⁴⁻⁷ agrees with the increased absorption we measured for Au NPs at 785 nm with rising temperature (Supplementary Fig. 23). And our data confirm that the relaxation of Au NPs slow down as the temperature increases (Supplementary Fig. 24). However, the shortened relaxation observed in Au-TiO₂ NRs with increasing temperature (Environmental heating) cannot be explained by the slowed relaxation predicted by the above models (Supplementary Figs. 22-24). The shorten τ_{e-ph} observed for Au-TiO₂ NRs at high environment temperature could be explained by the higher efficiency in hot carrier transfer, wherein accelerated relaxation via charge transfer aligns with previous experimental observations^{3,14}. And similar fast relaxation dynamics trend is also obtained for the winglet signal decay (Supplementary Fig. 25). Furthermore, the selective chemical enhancement observed in our TI-SERS (11.4-fold at 1,172 cm⁻¹, 2.8-fold at 907 cm⁻¹, Supplementary Table. 1) also serves as evidence for efficient charge transfer, as opposed to a non-selective enhancement^{8,10-13} expected from purely electromagnetic mechanism. Next, direct evidence for hot-hole transfer was provided by our EPR observations.”

2) We added this part in the main lines 381–383: “The sample was heated in situ using a Cindbest temperature-controlled stage equipped with a platinum resistance thermometer, which established a stable high-temperature environment through ambient thermalization.”

3) We added Supplementary Figs. 22–24.

Supplementary Fig. 22: The dynamics of laser heating. Dynamics of **a** Au NPs and **b** Au-TiO₂ NRs following laser energy densities of 160 $\mu\text{J cm}^{-2}$ and 90 $\mu\text{J cm}^{-2}$, extracted at the maximum winglet peak and pumped at 785 nm.

The relaxation dynamics slow down with increasing laser energy densities, consistent with previous standard “solid-state” integral model and experiments¹⁻³.

Supplementary Fig. 23: The in situ steady-state absorption spectra of **a** Au NPs and **b** Au-TiO₂ NRs, tested at room temperature, 120°C and 180°C; Differential-absorption spectra of **c** Au NPs and **d** Au-TiO₂ NRs relative to room-temperature absorbance, calculated from the data of **a** and **b**. The absorption of Au nanoparticles at 785 nm increases by ~40% at 180° C compared to 22° C, consistent with previous temperature-dependent permittivity models⁴⁻⁷.

Supplementary Fig. 24: Dynamics of Au NPs at 22°C and 180°C, extracted at the maximum winglet peak of Au, pumped at 785 nm. And Au NPs at 180°C exhibited slower τ_{e-ph} (9.7 ± 0.6 ps) than Au NPs at room temperature (8.6 ± 0.5 ps). Under 785 nm pumping, the lengthened relaxation of Au NPs at 180° C is likely a consequence of the increased photon absorption of Au at higher temperatures (Supplementary Fig. 23).

Q2. Furthermore, the mechanism itself is unclear - the authors emphasize that their system is described by a temperature, hence, the electrons should have a Fermi-Dirac distribution. In that context, what is meant by "hot-hole energy upconversion via absorbed phonons"? hot holes usually

refer to a deviation of the e system from Fermi-Dirac distribution (i.e., a non-thermal effect); the terminology implies the se charges are coming from the valence band, where under thermal conditions, their occupation should be completely negligible;” absorbed phonons” - where? by what? Overall, I do not think that the paper can be accepted before the above issues are seriously addressed. This may involve authors of a different expertise, but this will ensure that unsubstantiated mechanisms will not be proposed, and the results will be far more convincing.

Author Reply: Thank you for the reviewer’s comment. As the reviewer mentioned, the electrons should have a Fermi-Dirac distribution at high temperatures^{15,16}. In our text, the “hot holes” includes “hot nonthermal holes” (Those has not undergone carrier-carrier scattering) and “hot thermalized holes” (Experienced carrier-carrier scattering, but not experienced carrier-phonon scattering). As pointed out by the reviewer, the simultaneous absorption of phonons by hot holes is likely challenging due to the rapid relaxation of hot carriers (a timescale of hundreds of femtoseconds), in contrast to the long-lived excited states in semiconductors which allow phonon absorption leading to energy upconversion^{17,18} (Response Fig. 2). We have revised our interpretation and now propose a simpler mechanistic model. As shown in Fig. 4g, at high temperatures, thermal excitation induces the depletion of conduction band electrons near the Fermi level, thereby creating unoccupied states that facilitate the excitation of hot holes from higher energy levels. Thus, these hot holes at higher energy levels of Au NPs could transfer to TiO₂ (Fig. 4g). At room temperature, the photon energy of 785 nm (1.58 eV) is slightly below the threshold energy (1.6 eV) required for hot-hole transfer at Au-TiO₂ interface^{19,20}, rendering this process inhibited (Fig. 4h). The relevant discussions have been supplied into the main text lines 247–257.

Response Fig. 2: **a** Schematic of luminescence upconversion for laser cooling of semiconductors¹⁷. Ultrafast carrier longitudinal optical phonon collision aids efficient thermalization (2) of photogenerated cold carriers (1) and the sequential luminescence upconversion (3)¹⁷; **b** Superposition of photoluminescence spectra for 1.96 eV excitation (black) and 1.715 eV excitation (red), indicated by the red arrow. Inset: upconversion scheme of WSe₂¹⁸; **c** Schematic of the negatively charged and neutral exciton energy level diagram, showing the phonon-assisted upconversion process from X⁻ to X⁰¹⁸. The red curve is the charged exciton dispersion with its centre-of-mass momentum k_{-} ¹⁸.

Revision: 1) We have refined the title and related discussions of the manuscript. “Anomalous high-temperature induced surface-enhanced Raman properties via heat-assisted hot-hole transfer”

lines 65–70: “Experimental observations from time-resolved spectroscopy and electron paramagnetic resonance spectroscopy (EPR) firstly revealed the near-infrared driven hot-hole transfer from Au to TiO₂, beyond the only occurrences of electron transfer. The transfer of hot holes happened through environmental heating changes the electron distribution near the Fermi level, which benefit the transfer of hot electrons from Au to the molecule (Fig. 1a), thereby enhancing

Raman scattering.”

lines 294–297: “Environmental heating alters the electron distribution, leading to the formation of unoccupied states near the Fermi level, which realize the near-infrared photoexcitation of hot holes at high energy and facilitate their subsequent transfer from Au to TiO₂ (Fig. 4g).”

lines 316–319: “Our results suggest that, before photoexcitation, environmental heating changes the electron distribution near the Fermi level. Following photoexcitation, this distribution facilitates the excitation of hot holes at high energy levels.”

2) We added this part in the main lines 247–257: “Combined the above results, we conclude that high temperatures can trigger the photoexcitation of hot holes at higher energy levels, enabling their transfer. Specifically, the photon energy of 785 nm (1.58 eV) is slightly below the threshold energy (1.6 eV) required for hot-hole transfer at Au-TiO₂ interface^{19,20}. However, the work of pioneers has established that the high temperatures could alter the electron distribution near the Fermi level^{15,16}. This results in a consequent depletion of conduction band electrons near the Fermi level, thereby providing unoccupied states for the photoexcitation of hot holes at high energy levels (Fig. 4g). It indicates that the Au-TiO₂ NRs can achieve efficient near-infrared-driven hot hole transfer under high temperature. We term this phenomenon as heat-assisted near-infrared driven hot-hole transfer. This hot-hole transfer process is well known not permissible at room temperature (Fig. 4h).”

3) We replaced Fig. 4g and Fig. 4h.

Fig. 4 Schematic diagram of hot holes transfer from Au to TiO₂ at **g** high temperature and **h** room temperature.

Q3: The amplitude of the absorption signals for the excited state of Au”- what does that mean?

Author Reply: Thank you for the reviewer’s comment. We apologize for our negligence of the unclear expressions in the text. “The amplitude of the absorption signals for the excited state of Au” in previous versions described the transient absorption signal of Au increased with an increase in temperature. And this explanation has been included in the revised version lines 182–185 to ensure clear presentation.

Revision: lines 182–185: “As displayed in Fig. 4a–b, photoexcitation of the Au NPs LSPR results in a winglet signal and a bleach signal (centered at 491 and 678 nm, respectively)^{3,14}. The maximum amplitude of the winglet signal and bleach signal of Au considerably increase with increasing temperature (Fig. 4b), indicating a larger electron population on Au at high temperatures.”

Q4: fig. 2l- I expected to see numbers on the y-axis...

Author Reply: Thank you for the reviewer’s comment. We have added numbers on the y-axis of

Fig. 2l in the revised manuscript.

Revision:

Fig.2 I The absorption spectra of TiO₂ NRs and Au-TiO₂ NRs.

Q5. refs 2-4 have comments written on them. They are most likely all wrong, so should not be quoted.

Author Reply: Thanks for the reviewers' valuable suggestions. We have deleted refs 2-4 in the text.

Q6: Writing level and English grammar are on an insufficient level. MUST be improved.

Author Reply: Thanks to the reviewer for the useful suggestion, and we understand that clear and concise communication is crucial in scientific writing. In response, we have refined the language, paying close attention to sentence structure, word choice, and tone. Furthermore, we have polished the language by the editing service of Springer Nature to improve readability of the manuscript, the certificate is shown in Response Fig. 3. We appreciate for reviewers' warm work earnestly and hope that the correction will meet with approval.

Response Fig. 3: Certificate of English Language Editing.

Response to the comments of Reviewer 2:

Reviewer #2: Zhang et al. present a new SERS substrate, composed of photodeposited gold nanoparticles on TiO₂ nanorods. This substrate shows anomalous, and yet unobserved, high temperature SERS performance, with an optimum at 180°C. The substrate is used for the detection of various analytes, and the researchers attempt to explain the mode of action. The observation of high temperature SERS is, without doubt, an important finding that warrants rapid and prestigious publication. Based on the data presented by the authors, I have no doubt that this effect occurs in a stable and reproducible manner, and that it is highly useful for the detection of very small amounts of analytes.

Author Reply: We gratefully thank the reviewer for the positive comments and kind recommendation for the publication of our manuscript.

Q1. It is not clear which crystal phase dominates in the TiO₂ NRs. The authors compare the XRD pattern of their sample to the reference pattern of rutile, but the peaks do not match well. The peaks also do not match well with anatase. In addition, the small peaks near 26, 34, 38 and 52° are likely to result from the underlying FTO substate, which should be clearly marked. The authors instead conclude the presence of rutile from the TEM lattice fringes, but is this the only option? Or is it possible that the tubes are a mix of rutile and anatase, or that they are not well crystalline at all?

Author Reply: Thanks for the reviewer's suggestions. In the revised manuscript, to further clarify the crystal phase of the TiO₂ NRs, we measured their Raman spectrum (Supplementary Fig. 3). The results indicate that the synthesized TiO₂ NRs are of the rutile phase. Furthermore, we performed a more detailed measurement of the XRD data by reducing the scan rate from 10°/min to 2°/min. The results (Fig. 2j) show that the (101), (111), (211), (002), (301), and (112) crystal planes of the TiO₂ NRs match well with the reference pattern of rutile TiO₂. Combining the results of Raman, XRD and TEM results, the crystal phase dominates in the TiO₂ NRs is rutile (main lines 97 - 101). In addition, as suggested by the reviewer, we have marked the diffraction peaks originating from the FTO substrate (Fig. 2j and Supplementary Fig. 16b).

Revision: 1) We updated the XRD pattern in Fig. 2j and Supplementary Fig. 16b.

Fig. 2j XRD patterns of TiO₂ NRs and Au-TiO₂ NRs substrates.

Supplementary Fig. 16 **b** XRD patterns of the prepared Au-TiO₂ NRs after heating at 180°C for 70 min, measured at room temperature.

- 2) We added this part in the main text lines 97–101: “The XRD pattern indicates that the diffraction peaks of TiO₂ NRs are following the rutile phase (JCPDS No. 21-1276, Fig. 2j), and the Raman signal of TiO₂ NRs also exhibits distinct characteristics of the rutile phase (Supplementary Fig. 3), collectively confirming the formation of TiO₂ NRs in the rutile crystal structure.”
- 3) We added Supplementary Fig. 3.

Supplementary Fig. 3: Raman spectrum of our TiO₂ NR substrates as well as anatase TiO₂ powder (aladdin 99.9%) and rutile TiO₂ powder (aladdin 99.9%). Four major peaks that represent A_{1g}, E_g, the second-order effect (SOE), and B_{1g} Raman-active vibrational modes are located at 609, 446, 235, and 141 cm⁻¹ respectively, indicating that the predominant phase of the TiO₂ NRs is rutile.

Q2. I have doubts about the accuracy of fit in Figure S17. All fits are located significantly above most measurement points. It is expected that by statistical error, in case of a good fit, about half of all measurement points are located above the fitting curve and the other half below it. This is not the case here. Can you please comment how the fit was obtained, and why it was done in this way? Furthermore, is a decrease by ~half a picosecond actually a "significant decrease", as the authors mention? Or would it be possible that, for example, different portions of the same sample show a similar deviation? In a similar manner, though the fit in Figure 4c looks much more reliable, is the difference in relaxation times extracted from the fit statistically significant?

Author Reply: Thank you for the reviewer’s comment. We are sorry that the signal-to-noise ratio of the TAS plots in previous submission was poor, which led to suboptimal accuracy in the lifetime fitting process. We have re-conducted the TAS experiments and replaced the original TAS spectra

with higher-quality ones in the revised version (Figs. 4a–c and Supplementary Figs. 21 and 25). We re-fit the dynamics referred to the previous works^{14,21} (*Nat. Commun.*, 2024, 15(1); *Nat. Commun.*, 2022, 13(1)), and provided an explanation of the fitting method in the main texts (lines 187–192, and lines 383–388). The electronic dynamics of gold nanoparticles in the metallic state can be effectively described using the widely accepted two-temperature model²². This model accounts for the nonequilibrium energy transfer processes that occur following laser excitation. Upon ultrafast photoexcitation, conduction electrons in gold absorb photons and quickly thermalize via electron – electron scattering, reaching an electron temperature much higher than that of the lattice. These hot electrons then transfer energy to the lattice through electron – phonon coupling, causing lattice heating. Finally, the heated phonons undergo thermal diffusion through phonon – phonon coupling. According to the pervious works^{14,21}, the electron – phonon and phonon – phonon coupling can be well fitted with formula (1).

$$S_e(t) = IRF \otimes (A * e^{-\frac{t}{\tau_{e-ph}}} + B * e^{-\frac{t}{\tau_{ph-ph}}}) \quad (1)$$

As shown in Fig. 4c, the dynamics probed at bleach peak of Au-TiO₂ NRs at 180°C exhibited faster relaxation time constant (5.2 ± 0.3 ps) than at room temperature (6.7 ± 0.4 ps), and similar fast relaxation dynamics trend was also obtained on the winglet peak (Supplementary Fig. 25). Besides, the temperature-dependent experiments of TAS were totally in situ experiments by using a custom-made heating source to fix the samples (main texts lines 381–383), thereby avoiding the occurrence of similar deviations in different portions of the same sample. And we collected dynamics data from three randomly selected locations on Au-TiO₂ NRs (Response Fig. 4), demonstrating good consistency and thus also excluding the aforementioned deviations. Moreover, the original dynamic data show that even without fitting, Au-TiO₂ NRs at 180°C exhibits faster relaxation compared to at room temperature (Fig. 4c and Supplementary Fig. 25). And the fitting results are well matched with the original data and have a consistent trend, the obtained lifetime difference of 1.5 ps is also not within the error range. Therefore, the difference in relaxation times extracted from the fit is reliable.

Response Fig. 4 Dynamics from three different points (1, 2, 3) on the same Au-TiO₂ NRs substrates at 22°C, extracted at the maximum winglet peak of Au. The τ_{e-ph} of points 1, 2, and 3 are 6.6 ± 0.4 ps, 6.6 ± 0.4 ps, and 6.5 ± 0.4 ps respectively, demonstrating good consistency across the sample.

Revision: 1) We updated this part in the main text:

Fig. 4 TAS map of Au-TiO₂ NRs obtained at **a** 22°C and **b** 180°C upon 785 nm pump, pulse energy density of 160 μJ cm⁻²; **c** Dynamics of Au NPs at 22°C, Au-TiO₂ NRs at 22°C and 180°C, extracted at the maximum bleach peak of Au;

lines 187–192: “Signal fitting is performed using a double exponential decay convolved with the instrument response function. As illustrated in Fig. 4c, the dynamics of Au NPs exhibit a τ_{e-ph} (8.6 ± 0.5 ps) at room temperature (bleach signal). At room temperature, Au-TiO₂ NRs (bleach signal with blue circles) exhibit faster τ_{e-ph} (6.7 ± 0.4 ps) than Au NPs, which has been previously attributed to the electron transfer process from Au to TiO₂¹⁴. While the τ_{e-ph} (5.2 ± 0.3 ps) of Au-TiO₂ NRs further reduced with increasing temperature to 180°C.”

lines 381–383: “The sample was heated in situ using a Cindbest temperature-controlled stage equipped with a platinum resistance thermometer, which established a stable high-temperature environment through ambient thermalization.”

lines 383–388: “The instrument response function (IRF) is 200 fs. Electronic relaxation signals were deconvoluted by fitting with the following formula^{14,21}:

$$S_e(t) = IRF \otimes (A * e^{-\frac{t}{\tau_{e-ph}}} + B * e^{-\frac{t}{\tau_{ph-ph}}})$$

where $S_e(t)$ is the electron signal intensity with time delay. A and B , τ_{e-ph} and τ_{ph-ph} (phonon–phonon coupling time), are the amplitudes and time constants of the two exponential decay components.”

2) We updated this part in the supplementary information:

Supplementary Fig. 21: TAS data after excitation at 785 nm. Representative TAS map for **a** Au NPs and **b** TiO₂ NRs.

Supplementary Fig. 25: Dynamics of Au NPs at 22°C, Au-TiO₂ NRs at 22°C and 180°C, extracted at the maximum winglet peak of Au, pumped at 785 nm. Au NPs exhibit τ_{e-ph} of 8.6 ± 0.5 ps. Au-TiO₂ NRs at 180°C exhibited faster τ_{e-ph} (5.2 ± 0.3 ps) than that at room temperature (6.6 ± 0.4 ps).

Q3. Figure S18 shows that a Ti³⁺ signal is already visible in the dark, which is easily possible, given that laser irradiation for gold deposition may have slightly (photo)reduced the material. Since all EPR spectra are plotted in arbitrary units, it is not possible for the reader to decide whether more Ti³⁺ is present under irradiation. This is additionally made more complicated by the differences in temperature between the two measurements. This makes the "proof" for hot electron transfer appear weak. In addition, can the authors cite an article why the peak just below g=2 is assumed to belong to the Ti³⁺ signal? This is unexpected, given that the signal in Figure S18 displays exactly the same shape, yet without the additional small signal below g=2.

Author Reply: Thank you for the reviewer's comment. We have changed the vertical axis of the EPR spectra from arbitrary units to signal intensity and added the corresponding signal intensity values to the Fig. 4d and Supplementary Figs. 26–28. Based on the EPR spectra in Supplementary Fig. 26, we obtained the in situ EPR light-minus-dark difference spectra (i.e., the EPR spectra under light irradiation at the same temperature minus those under dark conditions) for Au-TiO₂ at different temperatures, as shown in Fig. 4d. The results indicate an increase in the h⁺(O⁻) and Ti³⁺ signals generated by light under high temperatures (90–180 °C). Additionally, the attribution of g=2 to Ti³⁺ may be questionable. We re-performed the EPR measurements on Au-TiO₂ NRs, including in situ EPR spectra under air and oxygen atmospheres, as well as spectra of heat-treated Au-TiO₂ (Supplementary Figs. 26–28). None of the spectra measured in the three experiments detected this small signal, and the relevant results have been replaced in the revised manuscript.

Revision: 1) We added this part in the main text lines 212–215: “On the basis of the EPR spectra in Supplementary Fig. 26, we obtained in situ EPR light-minus-dark difference spectra (i.e., the EPR spectra under light irradiation at the same temperature minus those under dark conditions) for Au-TiO₂ at different temperatures, as shown in Fig. 4d.”

Fig. 4: **d** In situ EPR light-minus-dark difference spectra of Au-TiO₂ NRs at different temperatures under 785 nm laser irradiation (the spectra were obtained in an air atmosphere).

2) We added this part in supplementary information.

Supplementary Fig. 26: The in situ EPR spectra of Au-TiO₂ NRs at different temperatures under **a** 785 nm laser irradiation and **b** dark conditions in the air atmospheres, and R22°C represents the in-situ cooling to 22°C.

Supplementary Fig. 27 **a** The decrease of Ti³⁺ on Au-TiO₂ NRs under 785 nm laser irradiation after adding oxygen at 180°C.

Supplementary Fig. 28: The comparison of EPR spectra of 180°C annealed (red) and unannealed (black) Au-TiO₂ NRs under 785 nm laser irradiation and dark (The spectra were obtained in an air atmosphere). The EPR spectra show the peak positions and signal amplitude of the Au-TiO₂ nanoarrays after heat treatment at 180 °C exhibit no significant changes compared with the untreated sample.

Q4. UV-Vis spectra must be re-recorded and displayed (raw data) in the Supplementary Information. The Tauc plot cannot be performed, the data is not clear. There is neither a clear baseline nor a clear absorption edge. Strictly speaking, the Tauc plot method is not appropriate anyway for composites of crystalline materials (see Klein et al., *Adv. Funct. Mater.* 33 (2023) 2304523). The reviewer understands, though, that the method is applied in comparison with previous works. Still, the band gap energy is estimated from the intercept of the tangent to the absorption edge with the baseline of the spectrum (not a random zero line), and neither of the two features (absorption edge, baseline) is clearly visible, as outline above.

Author Reply: Thank you for the reviewer’s comment. We fully agree with your concern that the UV-Vis spectra and Tauc plot are not clear. To enable comparison with previous works, we have re-recorded the UV-Vis spectra in Fig 21 and Supplementary Fig. 20, and the corrected Tauc plot result with distinct absorption edge and baseline of the spectra is shown in Response Fig. 5. According to the reviewer’s suggestion, we have revised the baseline of the spectra through linear fitting of the low-energy data points (Supplementary Note 5), referred to the previous works^{23,24}(*Nat. Energy*, 2022, 7(12); *J. Phys. Chem. Lett.*, 2018, 9(23)). As displayed in Response Fig. 5, the band gap values of TiO₂ NRs and Au-TiO₂ NRs obtained using the Tauc plots are 3.16 eV and 3.10 eV, respectively. The obtained band gap values for the TiO₂ NRs from Tauc plots show good consistency with the reported results obtained through the first-principles calculations²⁵ (*J. Semicond.* 2019, 40, (9)).

Response Fig. 5 The Tauc plots for the TiO₂ NRs and Au-TiO₂ NRs, respectively.

Revision: 1) we have adjusted the Fig. 21 and Supplementary Fig. 20.

Fig. 21 The absorption spectra of TiO₂ NRs and Au-TiO₂ NRs.

Supplementary Fig. 20: **a** The absorption spectra of the unannealed Au-TiO₂ NRs tested at room temperature (unannealed, RT) and at 180°C (unannealed, 180°C), as well as the 180°C-annealed Au-TiO₂ NRs tested at room temperature (annealed 180°C, RT); and **b** The corresponding Tauc plot analysis for band gap width.

2) We corrected the Tauc plot and corresponding discussion in Supplementary Note 5.

Supplementary Note 5. Data analysis of in situ absorption spectra from 22° C to 180° C on Au-TiO₂ NRs.

The band gap of Au-TiO₂ NRs was obtained from Tauc plots (Supplementary Fig. 20), which could be determined using the plots of $(\alpha hv)^2$ versus photon energy ($h\nu$) due to the direct gap of TiO₂²⁶. The baseline of the spectra was obtained through linear fitting of the low-energy data points^{23,24}, and the band gap values are estimated from the intercept of the tangent to the absorption edge with the baseline of the spectra. The absorption spectra and bandgap during and after heating remained consistent with the pristine Au-TiO₂ NRs, indicating that the heat treatment has no influence on the energy band structure of our substrates.

Q5. Considering points 1 and 3 above, it seems possible that the unexpected properties do not arise predominantly from an absorption by gold, but by titania. It is well known that reduced titania is blue, and that it shows broad absorption across the red and infrared range. The color of titania would be obscured, however, by the purple color due to the deposited gold. The alternative hypothesis of an action of reduced titania might be ruled out, for example, by an oxidative treatment of the final

(gold deposited) substrate, for example at 180°C (where it is stable) with high amounts of oxygen in the surrounding atmosphere. If the samples change color, and the SERS performance at 180° is lost or strongly attenuated, then the mechanism should potentially take into account a contribution of the reduced (and possibly amorphous) TiO₂.

Author Reply: We would like to thank the referee for their suggestion to further explore the role of the reduced TiO₂ in the origin of TI-SERS as a way to strengthen the rationality of this manuscript. According to the reviewer's suggestion, we have supplemented the TI-SERS spectra of Au-TiO₂ NRs with temperature increased from 22 to 180°C in the O₂ atmosphere. Under the oxygen atmosphere, the Raman signal of CV molecules notably increased as the temperature rose from 90° C to 180° C, and the Raman peak position and signal intensity were consistent with the results obtained in air (Supplementary Fig. 27). Besides, we further investigated the contribution of Ti³⁺ using in-situ temperature-dependent EPR spectra in the O₂ atmosphere at 180°C. Under the oxygen atmosphere, the signal of Ti³⁺ decreased compared to that in air, while the TI-SERS behavior is not affected by oxygen (Supplementary Fig. 27), indicating TI-SERS is not caused by the reduced TiO₂. Furthermore, we heated Au-TiO₂ NRs to 180° C in the air and oxygen atmosphere to study the color change of the sample (Response Fig. 6), while no obvious color changes were observed on the surfaces of Au-TiO₂ NRs (maintain purple-black color) compared to room temperature.

Response Fig. 6: A schematic illustration with photographs for the color changes of **a** Au-TiO₂ NRs substrates at 180° C in the air and oxygen atmosphere respectively; **b** A schematic illustration with photographs for the synthesis of Pt-DSA/TiO₂ through the photochemical defect tuning process, and the reduced titania shows as blue

color²⁷.

Revision: 1) We added this part in the main text lines 230–233: “Moreover, the Ti^{3+} signal also tend to increase as the temperature increases from 90°C to 180°C (Fig. 4d). We further confirmed that the decrease in Ti^{3+} does not affect the performance of TI-SERS through the EPR and Raman results in an oxygen atmosphere (Supplementary Fig. 27), thereby ruling out the contribution of Ti^{3+} to TI-SERS behavior.”

2) We added the new Supplementary Figs. 27 in the supplementary information.

Supplementary Fig. 27 **a** The decrease of Ti^{3+} on Au-TiO₂ NRs under 785 nm laser irradiation after adding oxygen at 180°C; **b** The TI-SERS spectra recorded from CV (10^{-7} M) on Au-TiO₂ NRs with temperature increased from 22 to 180°C in the O₂ atmosphere. An external mechanical pump and mass flow controllers were used to regulate the flow rates of O₂ at 40 ml/min to ensure the oxygen atmosphere; **c** The intensity comparison of the corresponding different Raman signal peaks at 180°C in the air and oxygen atmosphere.

Response to the comments of Reviewer 3:

The paper outlines work on SERS in high-temperature environments. The authors reported that Au-TiO₂ nanoarrays showed Raman signal enhancement with an increase in temperature. With a signal intensity increase at 180°C compared to that at 22°C. The paper ascribes this effect to phonon-assisted hot-hole energy upconversion. The paper showing that the increases in SERS signal, is probe molecule sensitive (Table S1 supplementary information). The substrate consists of quasi-oriented nanotubes+silver nano particles (see Fig 4 supplementary information). The authors reporting that the sample is polycrystalline. Ag nanoparticles are grown on the TiO₂ through UV (375 nm) laser irradiation. TI-SERS (needs to be define, but presumably meaning temperature induced SERS) studies from room temp to 180 degree C where undertaken. Before the distribution of the SERS studies the paper outlines characterization of the sample.

Author Reply: We gratefully thank the reviewer for the positive comments and kind recommendation for our manuscript. According to the reviewer's suggestion, we have clearly defined TI-SERS where it first appears in the revised manuscript ("high-temperature induced SERS" (TI-SERS), line 64).

Q1: Can information using methods be applied to determine in the impact of heating the substrate (upto 200 degrees), with regard to structure and electronic configuration (as seen via methods like XPS). Supplementary Fig. 7 shows that heating at c.a. 210 degrees is reported to course melting of the sample. A very important point of information is to clearly assess the impact of heating to 180 degrees on the sample. Does this cause any persevered changes noting its near the melting point.

Author Reply: Thank you for the reviewer's comment. According to the reviewer's suggestion, we have conducted in situ XPS analyses from 22 to 210° C to determine the impact of heating to 210° C on our substrates. Compared with 22° C, the proportion of oxygen species (O, V_O, and O_i) in the structure remain unchanged at 180° C but undergo changes at 200° C and 210° C (Response Fig. 7), especially the significant increase in oxygen incorporation (O_i). To assess the impact of heating to 180° C on the sample, we have performed XRD, in situ XPS and absorption spectra of sample from 22° C to 180° C (Supplementary Figs.16–20). The results show no notable change in the structure and electronic configuration when comparing samples at 180° C with room-temperature controls. The detailed analysis and description have been added in Supplementary Note 4.

Response Fig. 7 **a** In situ O 1s XPS spectra of the Au-TiO₂ NRs tested at room temperature (22°C), 180°C, 200°C, and 210°C; **b** The corresponding proportion of O, V_O, and O_i of the Au-TiO₂ NRs.

Revision: 1) We added Supplementary Figs.17–20 and Supplementary Note 4 in the supplementary information.

Supplementary Fig. 17: **a–b** XPS spectra of the unannealed Au-TiO₂ NRs tested at room temperature (unannealed, RT) and at 180°C (unannealed, 180°C), as well as the 180°C-annealed Au-TiO₂ NRs tested at room temperature (annealed 180°C, RT).

Supplementary Fig. 18: O 1s XPS spectra of the unannealed Au-TiO₂ NRs tested at room temperature (unannealed, RT) and at 180°C (unannealed, 180°C), as well as the 180°C-annealed Au-TiO₂ NRs tested at room temperature (annealed 180°C, RT).

Supplementary Fig. 19: Based on Supplementary Fig. 18, we obtained the proportion of O, V_O, and O_i in the

unannealed Au-TiO₂ NRs tested at room temperature (unannealed, RT) and at 180°C (unannealed, 180°C), as well as the 180°C-annealed Au-TiO₂ NRs tested at room temperature (annealed 180°C, RT).

Supplementary Fig. 20: **a** The absorption spectra of the unannealed Au-TiO₂ NRs tested at room temperature (unannealed, RT) and at 180°C (unannealed, 180°C), as well as the 180°C-annealed Au-TiO₂ NRs tested at room temperature (annealed 180°C, RT); and **b** The corresponding Tauc plot analysis for band gap width.

Supplementary Note 4. Data analysis of in situ XPS from 22° C to 180° C on Au-TiO₂ NRs.

The in situ XPS analysis of Au-TiO₂ NRs at 180°C (unannealed, 180°C) was conducted to investigate the changes of the sample during heating, and the XPS test at room temperature of the Au-TiO₂ NRs annealed at 180°C (annealed 180°C, RT) was applied to reflect the changes of the samples after heating. As shown in Supplementary Fig. 17, the binding energies of Au 4f and Ti 2p show no shift during and after heating compared to the pristine Au-TiO₂ NRs (unannealed, 180°C). What's more, the oxygen defects (oxygen vacancies and incorporation) may influence the charge transfer and band structure of TiO₂ heterojunctions^{13,28,29}. Therefore, the influences of heat treatment on chemical states of O 1s were characterized by XPS. As shown in Supplementary Figs. 18 and 19, the XPS pattern of O 1s can be deconvoluted into three peaks: 529.8 eV assigned to oxygen atoms in TiO₂ (O), 531.2 assigned to the oxygen vacancies (V_O), and 532.4 eV assigned to the additional O species adsorbed on the surface of the sample (O_i). It is obviously that the contents of O, V_O, and O_i remained unchanged during and after heating (Supplementary Figs. 18–19) compared to the pristine Au-TiO₂ NRs, which demonstrates that 180°C heat treatment cannot induce oxygen-related defects in our substrate.

Q2: The use of UV light on TiO₂ has been reported to cause the PIERS effect, similarly the heating of semiconductors like TiO₂ also causes such effects as shown via heating semiconductors including TiO₂ to c.a. 200 degrees (see Nature Communications volume 7, Article number: 12189 (2016), Journal of Physical Chemistry C 124 (46), 25351-25360 2020 or ACS Applied Nano Materials 3 (2), 1666-1673, 2020). Significantly this extends to enhancing plasmon assisted catalysis. The author in Fig 5 showing that heating the substrate supports catalysis reactivity. Reports have shown that heating Ag/TiO₂ and other semiconductors (Journal of Physical Chemistry C 124 (46)) supports not only an increase in SERS but also in catalysis.

Author Reply: Thanks for the reviewer's suggestions. The three works mentioned by the reviewers all focused on the improved room temperature SERS and photocatalytic performance, caused by generating oxygen defects (oxygen vacancies or oxygen incorporation) in the semiconductor after heat treatment or UV light^{13,28,29}. Our work concentrated on the enhanced high temperature SERS

and photocatalytic performance by in situ heating Au-TiO₂ NRs. In detail, when heated at 180°C, the XPS spectra (Response Fig. 8) revealed the relative concentration of O_i exhibits a variation of less than 1%, and absorption spectra (Response Fig. 9) showed the band gap changes by less than 0.1 eV. The results are differed from the work of oxygen incorporation (Journal of Physical Chemistry C 124 (46), 25351-25360 2020) that exhibits a significant increase in O_i content (more than 10%) and decrease in bandgap (more than 0.9 eV) after heated. It rules out oxygen incorporation as the underlying mechanism for TI-SERS.

Response Fig. 8 **a** O 1s XPS spectra of the unannealed Au-TiO₂ NRs tested at room temperature (RT) and 180°C (in situ), and the 180°C annealed Au-TiO₂ NRs tested at room temperature, **b** O 1s XPS spectra of unannealed and 200° C annealed WO₃ nanowires, both of them tested at room temperature²⁹.

Response Fig. 9 **a** The absorption spectra of the unannealed Au-TiO₂ NRs tested at room temperature (Unannealed (RT)) and 180°C (Unannealed (180°C)); **b** The corresponding Tauc plot analysis for band gap width; **c** The

normalized absorption spectra of unannealed and annealed WO₃/AgNPs substrates²⁹, and **d** The corresponding Tauc plots used to determine the indirect bandgap value²⁹.

Meanwhile, at room temperature, Raman spectra (Response Fig. 10) analysis revealed no enhancement in signal intensity for the 180° C-annealed samples when compared to the untreated control samples, differing significantly from the 2.5 fold enhancement achieved in the oxygen incorporation works (ACS Applied Nano Materials 3 (2), 1666-1673, 2020).

Response Fig. 10 **a** At room temperature, Raman spectra of 10⁻⁷ M CV on unannealed and 180° C annealed Au-TiO₂ NRs; **b** At room temperature, Raman spectra of 10⁻⁴ M MB on Ag NPs on non-treated and annealed at 200° C TiO₂²⁸.

What's more, the absorption spectra (Response Fig. 11) in our work show no LSPR peak shifts for Au-TiO₂ NRs during and after heating, which is different from the blue shift of LSPR peak caused by oxygen vacancies in UV light works (see Nature Communications volume 7, Article number: 12189 (2016)). These results demonstrate that the mechanism of TI-SERS is fundamentally distinct from those oxygen-defect-mediated processes reported in previous studies.

Response Fig. 11 **a** Absorption spectra show no LSPR shifts for Au-TiO₂ NRs at 180°C compared with at room temperature; **b** Absorption spectra of a thin film showing LSPR shifts for AuNPs on TiO₂ (R) after irradiation with 254nm light (3 h)¹³.

Revision: We added this part in the main text lines 168–176: “The mechanism of action for TI-SERS remains unclear. On the one hand, some previous studies have shown that oxygen-related

defects caused by high temperatures may enhance the Raman signal^{13,28,29}. To verify whether TI-SERS is related to oxygen defects generated under high temperature conditions, we conduct a series of experiments, such as XPS, XRD, EPR, Raman and absorption spectra (Supplementary Figs. 16–20 and 28, and Supplementary Notes 4–5). The results indicate that, compared with room temperature, heating the Au-TiO₂ NRs substrates to 180°C does not induce measurable changes in their structure or electronic configuration. This observation suggests that no oxygen defects are generated in the Au-TiO₂ NRs upon heating to 180 °C, indicating that the TI-SERS mechanism is not attributable to oxygen defects.”

Q3: Overall. The noteworthy results are a demonstration that SERS at high temperatures in Ag/TiO₂ systems can be improved, by an amount depending upon the probe molecule. More work in regard to assessing the impact of thermal treatment of the substrate is requested. Such as taking the substrate and examining the improvements in SERS at room temperature using the oven treated substrate (as outlined in Journal of Physical Chemistry C 124 (46), 25351-25360 2020 or ACS Applied Nano Materials 3 (2), 1666-1673, 2020).

Author Reply: Thank you for the reviewer’s comment. According to the reviewer’s suggestion, we have added this experiment in Supplementary Fig. 9c, which involved taking the substrate and examining the improvements in SERS at room temperature using the oven treated substrate (180 °C annealed for 70 min), following the previous studies (Journal of Physical Chemistry C 124 (46), 25351-25360 2020 or ACS Applied Nano Materials 3 (2), 1666-1673, 2020). While, the results indicated no SERS enhancement was observed at the 180°C annealed Au-TiO₂ NRs compared to the unannealed samples (see **Q2**, Response Fig. 10).

Revision: 1) We added this part in the main text lines 139–140: “Also, room-temperature SERS testing of the 180°C-annealed Au-TiO₂ NRs showed no detectable enhancement (Supplementary Fig. 9).”

2) In SI, we add Supplementary Fig. 9c.

Supplementary Fig. 9 c The Au-TiO₂ NRs were annealed at 180°C in an oven for 70 min. At room temperature, Raman spectra (10⁻⁷ M CV) revealed no SERS enhancement was observed for the 180°C-annealed Au-TiO₂ NRs (annealed 180°C, RT) compared with the pristine samples (unannealed, RT).

Q4: A careful analysis of the Ag/TiO₂ substrate following heat treatment is required. Eg after heating at room temperature what is the ESR data and what are the changes in structure, if any. Does the

presence of thermally induced defect states explain the observed ESR data for example.

Author Reply: Thank you for the reviewer's comment. We have added the room-temperature EPR data of 180 °C treated Au-TiO₂ NRs substrates in Supplementary Fig. 28, which show that the EPR signal of the Au-TiO₂ NRs after heat treatment at 180 °C exhibit no changes compared with original sample. The EPR data do not support the contribution of thermally induced defects to TI-SERS. In summary, the experimental findings demonstrate that oxygen defects are insufficient to explain TI-SERS, whereas our phonon-assisted hot hole energy upconversion mechanism provides a consistent explanation for the observed TI-SERS enhancement.

Revision: 1) In SI, we added Supplementary Fig. 28.

Supplementary Fig. 28: The comparison of EPR spectra of 180°C annealed (red) and unannealed (black) Au-TiO₂ NRs under 785 nm laser irradiation and dark (The spectra were obtained in an air atmosphere). The EPR spectra show the peak positions and signal amplitude of the Au-TiO₂ nanoarrays after heat treatment at 180 °C exhibit no significant changes compared with the untreated sample.

References

- 1 Del Fatti, N. *et al.* Nonequilibrium electron dynamics in noble metals. *Phys. Rev. B*, **61**, 16956 (2000).
- 2 Saavedra, J. R. M., Asenjo-Garcia, A. & García de Abajo, F. J. Hot-electron dynamics and thermalization in small metallic nanoparticles. *ACS Photonics*, **3**, 1637-1646 (2016).
- 3 Tagliabue, G. *et al.* Ultrafast hot-hole injection modifies hot-electron dynamics in Au/p-GaN heterostructures. *Nat. Mater.*, **19**, 1312-1318 (2020).
- 4 Reddy, H. *et al.* Temperature-dependent optical properties of single crystalline and polycrystalline silver thin films. *ACS Photonics*, **4**, 1083-1091 (2017).
- 5 Shen, P. *et al.* Temperature- and roughness-dependent permittivity of annealed/unannealed gold films. *Opt. Express*, **24**, 19254-19263 (2016).
- 6 Gurwich, I. & Sivan, Y. Metal nanospheres under intense continuous-wave illumination: A unique case of nonperturbative nonlinear nanophotonics. *Phys. Rev. E*, **96**, 012212 (2017).
- 7 Ferrera, M., Magnozzi, M., Bisio, F. & Canepa, M. Temperature-dependent permittivity of silver and implications for thermoplasmonics. *Phys. Rev. Mater.*, **3**, 105201 (2019).
- 8 Sun, G., Khurgin, J. B. & Bratkovsky, A. Coupled-mode theory of field enhancement in complex metal nanostructures. *Phys. Rev. B*, **84**, 045415 (2011).
- 9 Campion, A. & Kambhampati, P. Surface-enhanced Raman scattering. *Chem. Soc. Rev.*, **27**, 241-250 (1998).
- 10 Jiang, X. *et al.* Comparative study of semiconductor TiO₂ and noble metal Ag substrates: The differences between chemical enhancement and electromagnetic enhancement in SERS. *J. Raman Spectrosc.*, **49**, 1257-1264 (2018).
- 11 Schlücker, S. Surface-enhanced Raman spectroscopy: concepts and chemical applications. *Angew. Chem. Int. Ed.*, **53**, 4756-4795 (2014).
- 12 Han, X. X., Rodriguez, R. S., Haynes, C. L., Ozaki, Y. & Zhao, B. Surface-enhanced Raman spectroscopy. *Nat. Rev. Methods Primers*, **1**, 87 (2021).
- 13 Ben-Jaber, S. *et al.* Photo-induced enhanced Raman spectroscopy for universal ultra-trace detection of explosives, pollutants and biomolecules. *Nat. Commun.*, **7**, 12189 (2016).
- 14 Dey, A. *et al.* Hydrogen evolution with hot electrons on a plasmonic-molecular catalyst hybrid system. *Nat. Commun.*, **15**, 445-456 (2024).
- 15 Dubi, Y. & Sivan, Y. “Hot” electrons in metallic nanostructures—non-thermal carriers or heating? *Light:Sci. Appl.*, **8**, 89 (2019).
- 16 Ye, Z. *et al.* Phonon-assisted up-conversion photoluminescence of quantum dots. *Nat. Commun.*, **12**, 4283 (2021).
- 17 Li, J. Laser cooling of semiconductor quantum wells: Theoretical framework and strategy for deep optical refrigeration by luminescence upconversion. *Phys. Rev. B*, **75**, 155315 (2007).
- 18 Jones, A. M. *et al.* Excitonic luminescence upconversion in a two-dimensional semiconductor. *Nature Physics*, **12**, 323-327 (2016).
- 19 Huang, J. *et al.* Elucidating the origin of plasmon-generated hot holes in water oxidation. *ACS Nano*, **17**, 7813-7820 (2023).
- 20 Tagliabue, G. *et al.* Quantifying the role of surface plasmon excitation and hot carrier transport in plasmonic devices. *Nat. Commun.*, **9**, 3394 (2018).
- 21 Zhang, Q. *et al.* Simultaneous capturing phonon and electron dynamics in MXenes. *Nat. Commun.*, **13**, 7900-7909 (2022).

- 22 Zhou, M. *et al.* Evolution from the plasmon to exciton state in ligand-protected atomically precise gold nanoparticles. *Nat. Commun.*, **7**, 13240 (2016).
- 23 Ji, R. *et al.* Perovskite phase heterojunction solar cells. *Nat. Energy*, **7**, 1170-1179 (2022).
- 24 Makuła, P., Pacia, M. & Macyk, W. How to correctly determine the band gap energy of modified semiconductor photocatalysts based on UV - Vis spectra. *J. Phys. Chem. Lett.*, **9**, 6814-6817 (2018).
- 25 Cai, X., Zhang, P. & Wei, S.-H. Revisit of the band gaps of rutile SnO₂ and TiO₂: a first-principles study. *Journal of Semiconductors*, **40**, 092101 (2019).
- 26 Wang, L. *et al.* Enhancement of pyridine derivatives containing symmetrical substituents on the photocatalytic degradation of phenol and antibiotics by Er-Fe-TiO₂. *Chem. Eng. J.*, **410**, 128319 (2021).
- 27 Lee, C. W. *et al.* Photochemical tuning of dynamic defects for high-performance atomically dispersed catalysts. *Nat. Mater.*, **23**, 552-559 (2024).
- 28 Fularz, A., Almohammed, S. & Rice, J. H. Oxygen incorporation-induced SERS enhancement in silver nanoparticle-decorated ZnO nanowires. *ACS Appl. Nano Mater.*, **3**, 1666-1673 (2020).
- 29 Fularz, A., Almohammed, S. & Rice, J. H. Controlling plasmon-induced photocatalytic redox reactions on WO₃ nanowire/AgNPs substrates via defect engineering. *J. Phys. Chem. C*, **124**, 25351-25360 (2020).

Response to reviewer's comments

Reviewer 1 (Remarks to the Author):

The authors have answered the comments of the reviewers professionally, and improved their manuscript. I nevertheless still think that their proposed mechanism (now changed!) is not substantiated with existing theoretical knowledge. Specifically,

Author Reply: We gratefully thank the reviewer for the insightful comments and kind recommendation for the publication of our manuscript.

Q1. The authors indicate that in Au samples, the e-ph energy transfer rate slows down upon absorption/increase of temperature in their measurements and in previous work.

- They mention changes to absorption upon, but it is not clear why is this relevant.
- They do not explain the physical reason for the slowing down. I believe it is due to the increase of the heat capacity of the electrons with temperature. This is something that should be easy to quantify by using known temperature dependence of the electron heat capacity and the temperature change.

Author Reply: Thank you for your comment. As the reviewer mentioned, the reduction in the e-ph energy transfer rate is due to the increase in the electron heat capacity with the increase of temperature. According to the absorption spectra (Supplementary Fig. 24), the absorption of Au nanoparticles at 785 nm is enhanced at 180 °C compared with that at 22 °C. The stronger optical absorption leads to a higher electron temperature. Therefore, the electron–phonon energy relaxation at 180 °C become slower than those at 22 °C. We have further elaborated on the physical mechanism underlying the extended electron–phonon relaxation time for Au NPs at high temperatures in Supplementary Fig. 26 and Supplementary Note 6.

Revision: 1) Main text lines 200–202: “Our data confirm that the relaxation time of Au NPs lengthened as the temperature increases (Supplementary Fig. 24–25 and Supplementary Note 6).”.

2) We added Supplementary Fig. 26 and Supplementary Note 6.

Supplementary Fig. 26: **a** The heat capacity of the electrons as a function of electron temperature; **b** Two-temperature model simulations showing the temporal evolution of electron temperatures in Au NPs at different temperatures; **c** The corresponding normalized curves of the data in **b**.

Supplementary Note 6. Analysis of the lengthened τ_{e-ph} in Au NPs at higher temperatures.

The well-established two-temperature model¹⁻⁴ was used to rationalize the slow down relaxation of Au NPs at 180 °C (Supplementary Fig. 25), in which the energy exchange rate between electron and phonon follows:

$$C_e \frac{dT_e}{dt} = -G(T_e - T_l) \quad (\text{Supplementary Eqn. 2})$$

$$C_l \frac{dT_l}{dt} = G(T_e - T_l) \quad (\text{Supplementary Eqn. 3})$$

Where C_l ($2.5 \times 10^6 \text{ Jm}^{-3}\text{K}^{-1}$)³ and T_l are the heat capacity and temperature of lattice. And C_e and T_e are the heat capacity and temperature of electrons. For $T_e < 3000 \text{ K}$, their relationship is characterized by a linear dependence^{5,6}, $C_e = \gamma T_e$, where γ is the electronic heat capacity constant. Using a reported value for ($68 \text{ Jm}^{-3}\text{K}^{-2}$)³, we simulated the C_e - T_e relationship, as shown in Supplementary Fig. 26a. And G ($1.1 \times 10^{16} \text{ Wm}^{-3}\text{K}^{-1}$)⁷ is the electron-phonon coupling constant of Au. Based on the absorption spectra of the Au nanoparticles (A_{absorb} , Supplementary Fig. 24) and a constant 785 nm pump laser energy density (W_{pump}) at different ambient temperatures, the energy density absorbed by the Au nanoparticles (W_{absorb}) was calculated by $W_{\text{absorb}} = W_{\text{pump}} * A_{\text{absorb}}$, yielding values of $160 \mu\text{J}\cdot\text{cm}^{-2}$ at $22 \text{ }^\circ\text{C}$ and $222 \mu\text{J}\cdot\text{cm}^{-2}$ at $180 \text{ }^\circ\text{C}$. Applying the two-temperature model, the corresponding peak electron temperatures were estimated to be 1563 K at $22 \text{ }^\circ\text{C}$ and 1864 K at $180 \text{ }^\circ\text{C}$. We also used the two-temperature model to simulate the temporal evolution of electron temperature in Au NPs under different peak electron temperatures (Supplementary Fig. 26b and c). The simulations confirm that the electron-phonon relaxation prolonged by 1.5 ps at $180 \text{ }^\circ\text{C}$ compared to $22 \text{ }^\circ\text{C}$, which is in agreement with our experimental observations (Supplementary Fig. 25).

Q2. Then, the authors contrast their finding of a faster e-ph energy transfer rate in Au-TiO₂ samples. Assuming the heat capacity of electrons in that sample is the same, they need to suggest a stronger effect. They suggested that "thermal excitation induces the depletion of conduction band electrons near the Fermi level, thereby creating unoccupied states that facilitate the excitation of hot holes from higher energy levels." – From Fig. 4g, I suppose they mean instead something like - excitation of hot holes at energy levels deep below the Fermi energy of Au -. Either way, such a mechanism was suggested in the context of plasmonic catalysis, and is controversial; indeed famous studies of the Halas group using this combination of materials has already been shown to raise incorrect claims due to lack or erroneous control experiments. Therefore, the authors should strengthen their (now fairly speculative) claim by quantifying it. Specifically, one should be able to predict the change of occupation at the hole state and near the Fermi energy by using the formulation suggested already in the 1970's by Rosei et al.. Since the temperatures and temperature change are both much smaller than the Fermi energy of Au, I expect from earlier studies of transient permittivity studies (where heating can be more than 10 times stronger) that the impact of the hole population change to be much weaker than the e-ph changes observed for Au only samples by orders of magnitude, especially since the transitions the authors seem to be discussing are near the X point. These are known to contribute much less than transitions near the L point. A possible exception may have been reported in the context of photoluminescence, but if so, should come with a strong justification. Moreover, the authors should be able to quantify not only the change in the number of thermal/non-thermal holes, but also their tunnelling probability to the TiO₂, and then, relate the process to the experimental observations of the SERS changes. All these steps are not trivial, and I am pessimistic they may yield convincing conclusions... but the authors deserve a chance to demonstrate. some related work was done by the group of Andrea Baldi.

I therefore suggest to make a decision about the suitability of the manuscript to Nature comm.

Only after another revision.

Author Reply: Thank you for your comment. According to reviewer's suggestions, we quantified the change of the Fermi surface at 180 °C compared to 22 °C as 0.06 eV (Supplementary Note 7). In Au-TiO₂ NRs heterostructure, the interband transition thresholds of Au are calculated to be 1.55 eV at the X point and 1.60 eV at the Γ point (Supplementary Figs. 31). The energy of our laser is 1.58 eV. Therefore, 785 nm laser (1.58 eV) with the aid of 0.06 eV thermal energy could enhance the hot hole transfer from Au (Main text lines 233–236 and Supplementary Note 7).

We also discussed the quantum tunneling effect in the main text lines 240–243. The quantum tunneling effect could be negligible.

According to Monte Carlo modeling^{8,9}, we calculate that laser heating raises the lattice temperature by no more than 30 °C. In experiments, at an ambient temperature of 90 °C, the total lattice temperature under illumination does not exceed 120 °C, and EPR reflects a pronounced hole transfer. But, at an ambient temperature of 180 °C under dark conditions, no driven hole transfer is observed by EPR. These results indicate that photoinduced hole transfer rather than merely photothermal effects, aligning with Andrea Baldi et al's. model of hot carriers dominated chemical reaction^{8,9}.

Revision: 1) Main text lines 233–236: “Meanwhile, the computational investigations of Au-TiO₂ NRs indicate an increased probability of hot hole transfer from Au at high temperatures, with detailed analysis provided in Supplementary Figs. 31–32 and Supplementary Note 7. These computational results are consistent with the experimental findings.”

2) Main text lines 240–243: “Moreover, we discussed the temperature-related quantum tunneling effect¹⁰⁻¹⁵. The quantum tunneling probability in plasmonic metal heterostructures is commonly considered to be weak dependence on temperature¹²⁻¹⁵. In our TI-SERS and EPR experiments, the signals are strongly dependent on temperature, indicating that the quantum tunneling effect could be negligible.”

3) We added this part in the Supplementary information:

Supplementary Fig. 31: The calculated fatband structures for **a** Au component in the Au-TiO₂ NRs, and **b** Au bulk; The diameters of the circles represent the magnitude of density of states (DOS), and the Fermi level is set at 0 eV.

Supplementary Fig. 32: The calculated **a** band structures and **b** density of states for Au-TiO₂ NRs; The Fermi level is set at 0 eV.

4) We added this part in the Supplementary Note 7:

“As proposed by Rosei et al.¹⁶, on increasing the temperature, the region between the Fermi surface and the constant energy surface $E = -4K_B T$ experiences a decrease of electron population, thereby providing unoccupied states for the transition of low-energy d-band electrons. An increase in temperature from 22 °C to 180 °C ($\Delta T = 158$ °C) leads to an energy shift of $\Delta E = 0.06$ eV. Subsequently, to elucidate the mechanism of high-temperature-induced interfacial hot holes, we employed first-principles calculations of the Au-TiO₂ heterostructure interface and computed its fatband structures. The band structure of the Au component in the Au-TiO₂ heterostructure shows an interband transition threshold of approximately 1.55 eV near the X point (Supplementary Fig. 31a). Therefore, under 785 nm (1.58 eV) laser excitation, the energy is sufficient to trigger the interband transition at the X point. However, the fatband structure exhibits a low density of states at the X point, indicating a low transition probability. In contrast, the interband transition near the Γ point has a threshold of 1.60 eV coupled with a high electronic density of states. Thus, with the assistance of 0.06 eV thermal energy, the 785 nm laser excitation energy of 1.58 eV is sufficient (Supplementary Fig. 31a) to open the interband transition channel at the Γ point (1.60 eV), significantly enhancing the hot hole yield. Besides, we evaluated the change in the number of high-energy hot holes in d-band based on the fatband structures. We extracted the density of states values from the calculated results for transitions at -1.55 eV near the X point and in the range of -1.60 to -1.64 eV near the Γ point. The results indicate that the generation of high-energy hot holes could be enhanced by ~20 times at 180 °C compared to 22 °C.

We also calculated the band structure of pure Au, which revealed an interband transition threshold of approximately 1.80 eV near the X point (Supplementary Fig. 31b), in agreement with established literature¹⁷⁻¹⁹. The calculated interband transition energies of Au and the Au-TiO₂ heterostructure are 1.80 and 1.55 eV, indicating a 0.25 eV reduction for Au in the heterostructure. This reduction also benefits the generation of hot holes under 785 nm laser excitation.”

Reviewer 2 (Remarks to the Author):

The article is a revised version that I had reviewed previously. The authors have submitted a rebuttal. Indeed, the new work performed by the authors significantly increases the quality of the article. It seems now almost ready for publication, but a minor revision to clarify some additional questions is strongly advised:

Author Reply: We gratefully thank the reviewer for the positive comments and kind recommendation for the publication of our manuscript.

Q1. Why is the intense peak of the reference pattern of rutile ($\sim 27^\circ$, (100) peak) not observed in the diffraction pattern of the authors' sample? Does it have to do with the growth direction?

Author Reply: Thank you for your comment. The intense peak of the reference pattern of rutile TiO_2 at $\sim 27^\circ$ corresponds to the (110) peak²⁰⁻²⁴. As the reviewer mentioned, we did not observe the (110) peak in XRD pattern because the TiO_2 nanoarrays were almost vertically grown on the FTO surface along the (110) plane. In a standard XRD scan, the detectable peaks arise from lattice planes parallel to the surface of FTO. Consequently, the (110) peak, which is almost perpendicular to the surface of FTO, are usually absent in the XRD pattern²²⁻²⁴ (Response Fig. 1). To disrupt this vertical alignment, we removed TiO_2 NRs powder from the FTO substrate via intensive ultrasound for XRD testing, which showed the intense (110) peak of the reference pattern of rutile at $\sim 27^\circ$ (Supplementary Fig. 4).

Response Fig. 1: **a** SEM images of the TiO_2 nanorod arrays²²; **b** XRD patterns of TiO_2 , defective TiO_2 nanorod arrays (def- TiO_2), and Pt/def- TiO_2 nanorod arrays²². **c** SEM images of our TiO_2 NRs substrate; **d** XRD patterns of TiO_2 NRs and Au- TiO_2 NRs substrates.

Revision: 1) We added this part in the main lines 102–104: “The (110) plane of the TiO_2 NRs is

almost perpendicular to the substrate, thus is invisible in the XRD pattern (Fig. 2j, Supplementary Fig. 4).”

2) We added this part in the supplementary information:

Supplementary Fig. 4: XRD patterns of TiO₂ NRs powder. To confirm that the absence of the (110) peak was due to the vertical alignment, we removed TiO₂ NRs powder from the FTO substrate via intensive ultrasound for XRD testing, which showed the intense (110) peak of the reference pattern of rutile at ~27°.

Q2. Has a similar amount of sample, or even better, the exact same portion of the sample, been used in the EPR measurements? Otherwise, they are not quantitatively comparable.

Author Reply: Thank you for your comment. We used the exact same portion of the sample in the in situ EPR measurements, thus these obtained data are quantitatively comparable. And the experimental details were included in the main lines 408–415.

Revision: We added this part in the main lines 408–415: “The in situ EPR investigation was performed by a Bruker A300 spectrometer operating at the X-band frequency, which was equipped with an in situ digital temperature control systems ER4131VT and ER4141VT working in the temperature range of 100-600 K. The nanoarray sample was loaded into quartz glass tubes connected both to a high-vacuum pumping system and to a controlled gas feed (air or O₂). Irradiation was performed by a 785 nm laser (Leoptics) and the output radiation focused on the samples in the cavity by an optical fiber. The spectra were recorded before and after 10 min of 785 nm laser irradiation inside the EPR cavity at different temperatures.”

Reviewer 3 (Remarks to the Author):

The paper reports a novel high-temperature enhanced SERS (TI-SERS) effect using Au–TiO₂ nanorod arrays (NRs) under 785 nm excitation. Unlike traditional SERS, which usually weakens at elevated temperatures, this study shows a >11× enhancement at 180 °C. The authors attribute this to a heat-assisted hot-hole transfer mechanism from Au to TiO₂, which promotes hot-electron transfer to analyte molecules and thus enhances Raman scattering. The authors have responded in detail to the list of questions/issues from the reviewers. Overall I am happy that the paper be published. Noting the comments below.

Author Reply: We gratefully thank the reviewer for the positive comments and kind recommendation for the publication of our manuscript.

Q1: I would like to point out that the manuscript is technically sound but occasionally verbose and could benefit from editing for clarity. For example:

“Anomalous observed Raman signal enhancement” → “An anomalous enhancement of Raman signal was observed.”

Consistency in units (°C spacing, scientific notation, etc.) should be improved.

Author Reply: Thank you for your comment to improve the clarity and conciseness of our writing. We have carefully edited the entire manuscript, paying close attention to paragraph structure, sentence logic and tone. The original phrasing “firstly anomalous observed Raman signal enhancement with an increase in temperature” has been replaced with the more precise statement: “an anomalous enhancement of Raman signal with increasing temperature was observed for the first time.” (Main lines 27–28). All units have also been checked to ensure consistency throughout the manuscript, including the °C spacing, the scientific notation in Fig. 3e and Supplementary Fig. 13b, and the axis consistency in the TI-SERS plots (Fig. 5).

References

- 1 Groeneveld, R. H. M., Sprik, R. & Lagendijk, A. Femtosecond spectroscopy of electron-electron and electron-phonon energy relaxation in Ag and Au. *Phys. Rev. B*, **51**, 11433 (1995).
- 2 Zhou, M. *et al.* Evolution from the plasmon to exciton state in ligand-protected atomically precise gold nanoparticles. *Nat. Commun.*, **7**, 13240 (2016).
- 3 D, A. C. & Ray, A. Two-temperature model for ultrafast melting of Au-based bimetallic films interacting with single-pulse femtosecond laser: Theoretical study of damage threshold. *Phys. Rev. B*, **107**, 195402 (2023).
- 4 Hu, M. & Hartland, G. V. Heat dissipation for Au particles in aqueous solution: relaxation time versus size. *J. Phys. Chem. B*, **106**, 7029-7033 (2002).
- 5 Lin, Z., Zhigilei, L. V. & Celli, V. Electron-phonon coupling and electron heat capacity of metals under conditions of strong electron-phonon nonequilibrium. *Phys. Rev. B*, **77**, 075133 (2008).
- 6 Fang, R., Wei, H., Li, Z. & Zhang, D. Improved two-temperature model including electron density of states effects for Au during femtosecond laser pulses. *Solid State Commun.*, **152**, 108-111 (2012).
- 7 Gudde, J *et. al.* Damage threshold dependence on electron-phonon coupling in Au and Ni films. *Appl. Surf. Sci.*, 127-128, 40-45 (1998).
- 8 Kamarudheen, R., Aalbers, G. J. W., Hamans, R. F., Kamp, L. P. J. & Baldi, A. Distinguishing among all possible activation mechanisms of a plasmon-driven chemical reaction. *ACS Energy Lett.*, **5**, 2605-2613 (2020).
- 9 Kamarudheen, R., Castellanos, G. W., Kamp, L. P. J., Clercx, H. J. H. & Baldi, A. Quantifying photothermal and hot charge carrier effects in plasmon-driven nanoparticle syntheses. *ACS Nano*, **12**, 8447-8455 (2018).
- 10 Goldberg, O., Meir, Y. & Dubi, Y. Vibration-assisted and vibration-hampered excitonic quantum transport. *J. Phys. Chem. Lett.*, **9**, 3143-3148 (2018).
- 11 Zerah Harush, E. & Dubi, Y. Do photosynthetic complexes use quantum coherence to increase their efficiency? Probably not. *Sci. Adv.*, **7**, eabc4631 (2021).
- 12 Tan, S. F. *et al.* Quantum plasmon resonances controlled by molecular tunnel junctions. *Science*, **343**, 1496-1499 (2014).
- 13 Xu, B. & Tao, N. J. Measurement of single-molecule resistance by repeated formation of molecular junctions. *Science*, **301**, 1221-1223 (2003).
- 14 Govorov, A. O., Zhang, H. & Gun'ko, Y. K. Theory of photoinjection of hot plasmonic carriers from metal nanostructures into semiconductors and surface molecules. *J. Phys. Chem. C*, **117**, 16616-16631 (2013).
- 15 Shiraishi, Y. *et al.* Quantum tunneling injection of hot electrons in Au/TiO₂ plasmonic photocatalysts. *Nanoscale*, **9**, 8349-8361 (2017).
- 16 Rosei, R. Temperature modulation of the optical transitions involving the fermi surface in Ag: Theory. *Phys. Rev. B*, **10**, 474-483 (1974).
- 17 Beversluis, M. R., Bouhelier, A. & Novotny, L. Continuum generation from single gold nanostructures through near-field mediated intraband transitions. *Phys. Rev. B*, **68**, 115433 (2003).
- 18 Brown, A. M., Sundararaman, R., Narang, P., Goddard, W. A., III & Atwater, H. A. Nonradiative plasmon decay and hot carrier dynamics: effects of phonons, surfaces, and geometry. *ACS Nano*, **10**, 957-966 (2016).

- 19 Karaman, C. O., Bykov, A. Y., Kiani, F., Tagliabue, G. & Zayats, A. V. Ultrafast hot-carrier dynamics in ultrathin monocrystalline gold. *Nat. Commun.*, **15**, 703 (2024).
- 20 Wang, X. H. *et al.* Pyrogenic iron(III)-doped TiO₂ nanopowders synthesized in RF thermal plasma: phase formation, defect structure, band gap, and magnetic properties. *J. Am. Chem. Soc.*, **127**, 10982-10990 (2005).
- 21 Holder, C. F. & Schaak, R. E. Tutorial on powder X-ray diffraction for characterizing nanoscale materials. *ACS Nano*, **13**, 7359-7365 (2019).
- 22 Tian, Z. *et al.* Selective photoelectrochemical oxidation of glucose to glucaric acid by single atom Pt decorated defective TiO₂. *Nat. Commun.*, **14**, 142 (2023).
- 23 Feng, X. *et al.* Vertically aligned single crystal TiO₂ nanowire arrays grown directly on transparent conducting oxide coated glass: synthesis details and applications. *Nano Lett.*, **8**, 3781-3786 (2008).
- 24 Zhou, X. *et al.* Synergistic cooperation of rutile TiO₂ {002}, {101}, and {110} facets for hydrogen sensing. *ACS Appl. Mater. Interfaces*, **10**, 28199-28209 (2018).

Response to reviewer's comments

Reviewer #1 (Remarks to the Author):

I read the response to previous round of comments. I find the answer of the authors confusing, and reflecting a lack of understanding of relevant basic concepts. I am either not able to follow their logic, or do not find it convincing qualitatively/quantitatively. Due to that, at this stage, I recommend to reject the paper. Specifically,

Author Reply: Thank you for your time and comments on our manuscript.

Q1. It is very unlikely that the absorption spectrum changed from 160 to 220 $\mu\text{J cm}^{-2}$ (upon heating from 300K to 450K) which is $\sim 40\%$. Indeed, the thermo-derivatives of the permittivity (and hence the absorption) were measured by several groups to be on the order of 1×10^{-3} - $1 \times 10^{-2} \text{ K}^{-1}$ (see works by Shalaev group, Chu group and others).

Author Reply: Thank you for your comment. The measured absorption variation of 40% from 295K to 453K in our study is reasonable and consistent with both the results from the Chu group¹ and the Shalaev group², as well as with relevant experimental data³. As mentioned by the reviewer, the thermo-derivatives of the permittivity are on the order of 1×10^{-3} to $1 \times 10^{-2} \text{ K}^{-1}$ (Response Fig. 1a). However, it must be considered that the imaginary part of gold's permittivity around 785 nm has a small value of ~ 1.1 (Response Fig. 1c). With this in mind, Chu group reported a variation of $\sim 23\%$ (Response Fig. 1b) in the imaginary permittivity from 300K to 450K at 800 nm, and the Shalaev group observed a change of $\sim 55\%$ from 296 K to 473 K around 785 nm. The absorption change in our experiments over a similar temperature interval (295 K to 453 K) is $\sim 40\%$ (Response Fig. 1d), which aligns well with the reported trends. Moreover, the increased absorption of our Au NPs at elevated temperatures is consistent with the behavior reported in prior experimental work³ (Response Fig. 2). The optical density (OD) of the Au NPs at 785 nm was measured to be 2 mOD at 293 K and 8 mOD at 460 K³ (Response Fig. 2). Using the formula for percent absorption (A): $A = (1 - 10^{-\text{OD}})$, the corresponding percent absorption are $\sim 0.5\%$ and $\sim 1.8\%$ respectively³. The absorption rose by 1.3%, representing a relative increase of $\sim 260\%$ ³.

Response Fig. 1: Temperature-dependent permittivity at selected wavelengths with gold films: **a** the thermo-derivatives of imaginary part ϵ''^1 , and **b** the corresponding ϵ''^1 ; **c** Temperature dependent imaginary part of the

dielectric function for gold films²; **d** Temperature-dependent differential absorption spectra of gold nanoparticles (Au NPs) plotted as the relative change in absorbance, defined as $[A(T)-A_{RT}]/A_{RT}$, where $A(T)$ is the percent absorption at elevated temperature T and A_{RT} is the absorbance at room temperature (22 °C).

ACS Photonics, 2020, 7(4), 959

Response Fig. 2: **a** Experimental static absorbance spectra of Au NPs at different temperature³; **b** Differential-absorbance spectra relative to room temperature absorbance, calculated from the data of panel **a**³.

Q2. I do not understand what is the meaning of the statement – “the electron-phonon relaxation prolonged by 1.5 ps at 180 °C compared to 22 °C”. what is the initial e-ph coupling time the authors have in mind? I would expect a change of decay time to be reported in percent. The value reported reflects an extremely large, unrealistic change, and raises an alarm about the whole argument. Does it even correlate to the experimental finding?

Author Reply: Thank you for your comment. To simulate the temperature-dependent electron-phonon relaxation of Au under constant pump fluence (Supplementary Fig. 25a), we simultaneously considered the effects of temperature-dependent absorbed energy (see Q3 and Q4) and lattice temperature. Applying the two-temperature model with constant pump fluence (Supplementary Fig. 26c), the e-ph coupling times (τ_{e-ph}) of Au NPs were determined to be 9.8 ps at 180 °C and 8.3 ps at 22 °C. This indicates that the τ_{e-ph} at 180 °C is 1.5 ps longer than that at 22 °C, representing a prolongation of ~18%.

Revision: 1) We have revised Supplementary Fig. 26:

Supplementary Fig. 26: **a** The heat capacity of the electrons as a function of electron temperature; **b** With constant

pump fluence, two-temperature model simulations showing the temporal evolution of electron temperatures in Au NPs at 22 °C and 180 °C; and **c** The corresponding normalized curves of the data in **b**; **d** With equal absorption energy, two-temperature model simulations showing the normalized temporal evolution of electron temperatures in Au NPs at 22 °C and 180 °C.

2) We revised this part to Supplementary Note 6: “With constant pump fluence, the simulated τ_{e-ph} of Au NPs was 9.8 ps at 180 °C and 8.3 ps at 22 °C (Supplementary Fig. 26c), representing a 18% increase. This trend agrees with our experimental observations (Supplementary Fig. 25a).”

Q3. Either way, quantifying the decay (cooling) rate should not be related to the absorption stage; thus, referring to it is quite surprising, and raises doubt about the whole line of thought of the authors, especially since the equations 2-3 in the SI do not involve absorption.

Author Reply: Thank you for your comment. The absorption of Au increases with temperature in experiments, which has been demonstrated by several groups¹⁻³ (see Q 1). The increased optical absorption typically leads to an increase in the electron-phonon coupling time τ_{e-ph} ^{4,5}. Therefore, to correspond with the experiments, we have taken absorption into account in the previous revision (NCOMMS-25-12718B). According to the optical density (OD) of the Au NPs (Supplementary Fig. 24a) and using the formula for percent absorption (A): $A = (1-10^{-OD})$, the percent absorption of Au NPs is 5.40% at 22 °C and 7.54% at 180 °C, respectively. Based on the relationship $E_{\text{absorb}}=E_{\text{pump}}\times A$, the absorbed energy for Au NPs at 785 nm are 0.32 μJ at 22 °C and 0.45 μJ at 180 °C, yielding the corresponding peak electron temperatures of 1563 K and 1864 K. Thus, the variation in absorption, as directly reflected in the change of the peak electron temperature, has been considered in our analysis.

Q4. Instead, I would have expected the authors to perform at least a crude estimate of the e-ph energy transfer time. A simple analysis reveals that what matters is not the heating level they refer to (which is similar for both cases, assuming the absorption does not change drastically), but rather the temperature itself. Such estimate should then be compared to the experimental measurement. No such analysis was done.

Author Reply: Thank you for your comment. Assuming equal absorption energy for Au at 22 °C and 180 °C, we applied the two-temperature model and obtained corresponding τ_{e-ph} of 8.3 ps at 22 °C and 9.0 ps at 180 °C (Supplementary Fig. 26d), indicating a slight slowdown at 180 °C. Furthermore, the absorbed energy by Au NPs (E_{Au}) in TAS is: $E_{\text{Au}}= E_{\text{pump}}\times (1-10^{-OD})$. By adjusting the pump laser intensity, we ensured that the Au NPs absorbed the equal energy at both 22 °C and 180 °C, and obtained the TAS results (Supplementary Fig. 25b). The data show that the corresponding τ_{e-ph} of Au NPs is 9.1 ± 0.5 ps at 180 °C, which is slightly longer than the 8.6 ± 0.5 ps observed at 22 °C (Supplementary Fig. 25b). This slight slowdown aligns with both established temperature-dependent experimental observations⁶ (Response Fig. 3) and the results of our two-temperature model simulation (Supplementary Fig. 26d). More importantly, Au NPs is merely a control experiment to demonstrate that high temperatures do not shorten the τ_{e-ph} of Au, thereby confirming that the reduced τ_{e-ph} of Au in the Au-TiO₂ NRs at 180 °C is due to charge transfer.

Response Fig. 3: **a** Dynamics of Au NPs at 22 °C and 180 °C with equal absorption energy, and Au NPs at 180 °C exhibited slightly slower τ_{e-ph} (9.1 ± 0.5 ps) than Au NPs at 22 °C (8.6 ± 0.5 ps); All kinetics were extracted at the maximum winglet peak of Au, pumped at 785 nm. **b** Time-resolved measurements of the change of the reflectivity on Ag for various temperatures⁶. It is seen that the subpicosecond relaxation speeds up as the temperature decreases⁶.

Revision: 1) We added this part to Supplementary Note 6: “To isolate the influence of lattice temperature on the τ_{e-ph} in Au NPs, we performed supplementary dynamics experiments with equal absorption energy (Supplementary Fig. 25b). By adjusting the pump intensity, we obtained an equal absorption energy density of of $160 \mu\text{J cm}^{-2}$ across temperatures. The τ_{e-ph} for Au NPs exhibits a slight increase at 180 °C (9.1 ± 0.5 ps) relative to 22 °C (8.6 ± 0.5 ps), which aligns with established experimental observations⁶. With equal absorption energy, the two-temperature model yields τ_{e-ph} values of 8.3 ps at 22 °C and 9.0 ps at 180 °C (Supplementary Fig. 26d), a trend that agrees with our experimental observations.”

2) We have added Supplementary Fig. 25b:

Supplementary Fig. 25: **a** Dynamics of Au NPs at 22 °C and 180 °C with constant pump fluence, and Au NPs at 180 °C exhibited obviously slower τ_{e-ph} (9.7 ± 0.6 ps) than Au NPs at 22 °C (8.6 ± 0.5 ps); **b** Dynamics of Au NPs at 22 °C and 180 °C with equal absorption energy, and Au NPs at 180 °C exhibited slightly slower τ_{e-ph} (9.1 ± 0.5 ps) than Au NPs at 22 °C (8.6 ± 0.5 ps); All kinetics were extracted at the maximum winglet peak of Au, pumped at 785 nm.

Q5. The authors write – “we quantified the change of the Fermi surface at 180 °C compared to 22 °C as 0.06 eV (Supplementary Note 7).” This is unlikely - this is 3 times the temperature.

Author Reply: Thank you for your comment. The reviewer mentioned it in the second review (NCOMMS-25-12718A-Z): “Specifically, one should be able to predict the change of occupation at the hole state and near the Fermi energy by using the formulation suggested already in the 1970’s by Rosei et al.”. According to Rosei’s work (Response Fig. 4a), the electron population between the Fermi surface and the $E=E_F-4k_B T$ surface decreases as temperature rises. 180 °C corresponds to 453.2 K and 22 °C corresponds to 295.2 K. Thus, substituting the values into the expression yields $\Delta E=4\times(8.61710^{-5} \text{ eV/K})\times(453.2-295.2) \text{ K}=0.06 \text{ eV}$. Furthermore, we simulated the Fermi–Dirac distribution ($f(E)=1/(e^{\frac{E-E_F}{k_B T}}+1)$) for Au at 22 °C and 180 °C (Response Fig. 4b–c), which reveals a maximum ΔE of $\sim 0.17 \text{ eV}$ corresponding to complete occupation ($f(E)=1$). Therefore, $\Delta E=0.06 \text{ eV}$ employed in our analysis is a reasonable and appropriate value. Our analysis strictly follows the study you previously recommended. I am confused about the basis for your comment regarding “this is 3 times the temperature”.

Response Fig. 4: **a** Rosei’s definition of the energy region near the Fermi level in which electron population is affected by temperature⁷; **b** A comparison of the Fermi-Dirac distribution for Au at -273.15 °C, 22 °C, and 180 °C; **c** The enlarged view of the key region near Fermi level.

Q6. The authors refer to a transition of “1.60 eV at the Γ point (Supplementary Figs. 31)”. This transition is irrelevant for interband transitions in Au (because the Gamma point has many occupied states in the conduction band), and implies the authors did not understand or did not read the earlier work by Rosei, which is a well accepted benchmark.

Author Reply: Thank you for your comment. The comment is somewhat ambiguous, and we may not fully understand why the reviewer considers the 1.60 eV transition at the Γ point to be irrelevant. Nevertheless, we attempt to respond as follows. Our DFT results of the Au component in Au-TiO₂

NRs indicate that the transition from the d-band to the Fermi surface near the Γ point exhibits a threshold of 1.60 eV (Response Fig. 5a). As shown in Response Fig. 5a, this transitions near Γ are almost forbidden when excited with 1.58 eV laser, as the corresponding final states are fully occupied and lie below the Fermi level. However, According to Rosei's work ($\Delta E = -4k_B T$), there is a decrease in the electron population within $\Delta E = 0.06$ eV below the Fermi surface at 180 °C relative to 22 °C, which creating unoccupied final states in this region. Therefore, with the assistance of $\Delta E = 0.06$ eV, the excitation energy of 1.58 eV is sufficient to open the transition channel at the Γ point. Moreover, we plotted the Fermi-Dirac distributions of Au at 22 °C and 180 °C (Response Fig. 5b), and the corresponding enlarged view of $f(E)$ from 0.8 to 1.0 is shown in Response Fig. 5c. When $f(E) = 1$, the distribution shifts by -0.31 eV at 22 °C and by -0.48 eV at 180 °C relative to the Fermi level. This yields a $\Delta E = 0.17$ eV, indicating that the transition final-state energy at 180 °C can be lowered by 0.17 eV compared to that at 22 °C. Hence, at 180 °C, a photon energy of 1.58 eV is capable to driving transitions at the Γ point.

Response Fig. 5: **a** The calculated fatband structures for Au component in the Au-TiO₂ NRs, and the blue dashed box represents the initial state range where transitions can occur at high temperatures; **b** A comparison of the Fermi-Dirac distribution for Au at -273.15 °C, 22 °C, and 180 °C; **c** The enlarged view of the key region near Fermi level in **b**.

Revision: We have revised Supplementary Fig. 31a:

Supplementary Fig. 31: **The calculated fatband structures for Au component in the Au-TiO₂ NRs, and the blue dashed box represents the initial state range where transitions can occur at high temperatures; b** The calculated fatband structures for Au bulk; The diameters of the circles represent the magnitude of density of states (DOS).

Q7. The authors write – “Therefore, 785 nm laser (1.58 eV) with the aid of 0.06 eV thermal energy could ...” – this reflects a lack of understanding of how photon absorption works and generates non-thermal holes.

Author Reply: Thank you for your comment. Our mechanism was clearly understood by the other two reviewers. Furthermore, after we submitted the manuscript, a similar mechanism (Response Fig. 6) was confirmed by recent work on gold⁸ published in Nature Communications (2025, 16(1), 2274). Therefore, from an analytical perspective, we hold that the transition in our mechanism is achievable (see Q5).

Response Fig. 6: **a** Au NPs hot carrier generation, multiplication and thermalization were investigated using pump-probe ultrafast transient XANES with colloidal nanoparticles in water, circulated in a liquid jet. The expected changes due to optical excitation are schematically represented⁸; **b** Schematic diagram of hot holes transfer from Au to TiO₂ at high temperature.

References

- 1 Shen, P. *et al.* Temperature- and roughness-dependent permittivity of annealed/unannealed gold films. *Opt. Express*, **24**, 19254-19263 (2016).
- 2 Reddy, H., Guler, U., Kildishev, A. V., Boltasseva, A. & Shalaev, V. M. Temperature-dependent optical properties of gold thin films. *Opt. Mater. Express*, **6**, 2776-2802 (2016).
- 3 Ferrera, M. *et al.* Thermometric calibration of the ultrafast relaxation dynamics in plasmonic Au nanoparticles. *ACS Photonics*, **7**, 959-966 (2020).
- 4 Brown, A. M. *et al.* Experimental and ab initio ultrafast carrier dynamics in plasmonic nanoparticles. *Phys. Rev. Lett.*, **118**, 087401 (2017).
- 5 Tagliabue, G. *et al.* Ultrafast hot-hole injection modifies hot-electron dynamics in Au/p-GaN heterostructures. *Nat. Mater.*, **19**, 1312-1318 (2020).
- 6 Groeneveld, R. H. M., Sprik, R. & Lagendijk, A. Femtosecond spectroscopy of electron-electron and electron-phonon energy relaxation in Ag and Au. *Phys. Rev. B*, **51**, 11433 (1995).
- 7 Rosei, R. Temperature modulation of the optical transitions involving the fermi surface in Ag: Theory. *Phys. Rev. B*, **10**, 474-483 (1974).
- 8 Wach, A. *et al.* The dynamics of plasmon-induced hot carrier creation in colloidal gold. *Nat. Commun.*, **16**, 2274 (2025).

Anomalous high-temperature induced surface-enhanced Raman properties via phonon-assisted 2 hot-hole energy upconversion - referee report

Zhang *et al.*

The authors suggest Au-TiO₂ particles as a platform for high temperature SERS. They claim an enhancement of about an order of magnitude at about 100 degrees higher than before in a stable manner. The results may be technically useful, but I am not convinced by the mechanism proposed, since simpler explanations were not ruled out.

In particular, one of the main findings (based on TAS) is a reduction of the e-ph decay time between 4.3 ps to 3.7 ps upon heating to 180 C. This effect of 14% can be easily explained via a standard "solid-state" integral model of the effect, see, e.g., del Fatti et al., PRB 2000 (and countless re-uses of the approach for pulsed and CW illumination). There is also various simpler approaches (e.g., Grua et al., PRB 2003, Saavedra et al, ACS Photonics 2016). The effect may also be reflected via temperature-dependent permittivity models, e.g., by the Shalaev group (various ellipsometry studies for Au, Ag, ...), the Chu group (Scientific Reports 2016, Nanophotonics 2017, Opt. Exp. 2017), Gurwich et al. (PRE 2017), Ferrara et al. (Phys Rev Mat 2019), etc..

Such a simple electromagnetism-based explanations should be accompanied by a quantification of the SERS enhancement, a procedure which is well established (e.g., Greg Sun and Khurgin, or more sophisticated models that properly account for spatial variations of the field enhancement around the plasmonic structure.)

Furthermore, the mechanism itself is unclear - the authors emphasize that their system is described by a temperature, hence, the electrons should have a Fermi-Dirac distribution. In that context, what is meant by "hot-hole energy upconversion via absorbed phonons"? hot holes usually refer to a deviation of the e system from Fermi-Dirac distribution (i.e., a non-thermal effect); the terminology implies the se charges are coming from the valence band, where under thermal conditions, their occupation should be completely negligible; "absorbed phonons" - where? by what?

Overall, I do not think that the paper can be accepted before the above issues are seriously addressed. This may involve authors of a different expertise, but this will ensure

that unsubstantiated mechanisms will not be proposed, and the results will be far more convincing.

Specific comments:

- "The amplitude of the absorption signals for the excited state of Au" - what does that mean?
- fig. 21 - I expected to see numbers on the y-axis...
- refs 2-4 have comments written on them. They are most likely all wrong, so should not be quoted.
- Writing level and English grammar are on an insufficient level. MUST be improved.